# Topological data analysis distinguishes parameter regimes in the Anderson-Chaplain model of angiogenesis

**John T. Nardini**[1], **Bernadette J. Stolz**[2], **Kevin B. Flores**[1], **Heather A. Harrington**[2,3], **Helen M. Byrne**[2]*

**1** Department of Mathematics, North Carolina State University, Raleigh, North Carolina, United States of America, **2** Mathematical Institute, University of Oxford, Oxford, United Kingdom, **3** Wellcome Centre for Human Genetics, University of Oxford, Oxford, United Kingdom

* helen.byrne@maths.ox.ac.uk

**Data Availability Statement:** All data is available at https://github.com/johnnardini/Angio_TDA.

**Funding:** HAH, BJS and HMB are grateful for the support provided by the UK Centre for Topological

## Abstract

Angiogenesis is the process by which blood vessels form from pre-existing vessels. It plays a key role in many biological processes, including embryonic development and wound healing, and contributes to many diseases including cancer and rheumatoid arthritis. The structure of the resulting vessel networks determines their ability to deliver nutrients and remove waste products from biological tissues. Here we simulate the Anderson-Chaplain model of angiogenesis at different parameter values and quantify the vessel architectures of the resulting synthetic data. Specifically, we propose a topological data analysis (TDA) pipeline for systematic analysis of the model. TDA is a vibrant and relatively new field of computational mathematics for studying the shape of data. We compute topological and standard descriptors of model simulations generated by different parameter values. We show that TDA of model simulation data stratifies parameter space into regions with similar vessel morphology. The methodologies proposed here are widely applicable to other synthetic and experimental data including wound healing, development, and plant biology.

## Author summary

Vascular networks play a key role in many physiological processes, by delivering nutrition to, and removing waste from, biological tissues. In cancer, tumors stimulate the growth of new blood vessels, via a process called angiogenesis. The resulting vascular structure comprises many inter-connected vessels which lead to emergent morphologies that influence the rate of tumor growth and treatment efficacy. In this work, we consider several approaches to summarize the morphology of synthetic vascular networks generated from a mathematical model of tumor-induced angiogenesis. We find that a topological approach can be used quantify vascular morphology of model simulations and group the simulations into biologically interpretable clusters. This methodology may be useful for

Data Analysis EPSRC grant EP/R018472/1. HAH gratefully acknowledges funding from EPSRC EP/R005125/1 and EP/T001968/1. This work was partially supported by the National Science Foundation under Grant DMS-1638521 to the Statistical and Applied Mathematical Sciences Institute, and in part by National Institute of Aging grant R21AG059099 to KBF. The funders had no role in study design, data collection and analysis, decision to publish, or preparation of the manuscript.

**Competing interests:** The authors have declared that no competing interests exist.

the diagnosis of abnormal blood vessel networks and quantifying the efficacy of vascular-targeting treatments.

## Introduction

Blood vessels deliver nutrients to, and remove waste products from, tissues during many physiological processes, including embryonic development, wound healing, and cancer [1–3]. The structure and form of connected blood vessels (e.g., vascular morphology), determine how nutrients and waste are supplied to or removed from the environment and, in turn, influence the behavior of the tissue and its constituent cells. The morphology of a vascular network can reveal the presence of an underlying disease, or predict the response of a patient to treatment. High resolution vascular imaging technology creates an exciting opportunity to develop mathematical tools to discover new links between blood vessel structure and function.

Here, we focus on *tumor-induced angiogenesis*, the process by which tumor cells stimulate the formation of new blood vessels from pre-existing vasculature [1]. When oxygen and nutrient levels within a population of tumor cells become too low to sustain a viable cell population, the tumor cells produce several growth factors, including vascular endothelial growth factor (VEGF), platelet-derived growth factor (PDGF), and basic fibroblast growth factor (bFGF), which diffuse through the surrounding tissue [4–6]. On contact with neighboring blood vessels, these tumor angiogenesis factors (or TAFs) increase the permeability of the vessel walls and loosen adhesive bonds between the endothelial cells that line the blood vessels [7]. The TAFs activate the endothelial cells to release proteases that degrade the basement membrane [8]. Activated tip endothelial cells then migrate away from the parent vessel and follow external cues, such as spatial gradients of VEGF and/or fibronectin [9, 10]. Stalk endothelial cells located behind the tip cells proliferate into the surrounding tissue matrix, causing the emerging vessel sprouts to elongate [11]. When tip endothelial cells from one sprout come into contact with tip or stalk endothelial cells from another sprout, the two endothelial cells may fuse or anastomose together, forming a new loop through which oxygen and nutrients may be transported [9]. In Fig 1 we present a schematic model of how these processes are coordinated.

As the number of experimental and theoretical studies of angiogenesis has increased, our knowledge of the mechanisms involved in its regulation has increased beyond the idealized description given above [12, 13]. For example, subcellular signalling involving the VEGF and delta-notch signalling pathways is known to influence whether an endothelial cell adopts a tip or stalk cell phenotype [14]. Further, immune cells, including macrophages that are present in the tumor microenvironment, are known to release TAFs such as VEGF and matrix degrading proteases which facilitate tumor angiogenesis [12]. Mechanical stimuli also influence endothelial cell migration. For example, the fluid shear stress experienced by endothelial cells lining blood vessels drives their migration by causing them to form lamellipodia and focal adhesions at the front of the cell, in the flow direction, and to retract focal adhesions at the rear [15, 16].

In this paper, we explore several approaches for quantifying the morphology of vascular networks that form during angiogenesis, including recently developed techniques from topological data analysis. We develop and test our methodology through application to simulated data from a highly idealized, and well known hybrid (i.e., discrete and continuous) stochastic mathematical model of tumor-induced angiogenesis proposed by Anderson and Chaplain [17] (see Fig 1 for a model schematic).

The Anderson-Chaplain model of tumor angiogenesis is based on continuum models proposed by Balding and McElwain [18] and Byrne and Chaplain [19], as well as the stochastic

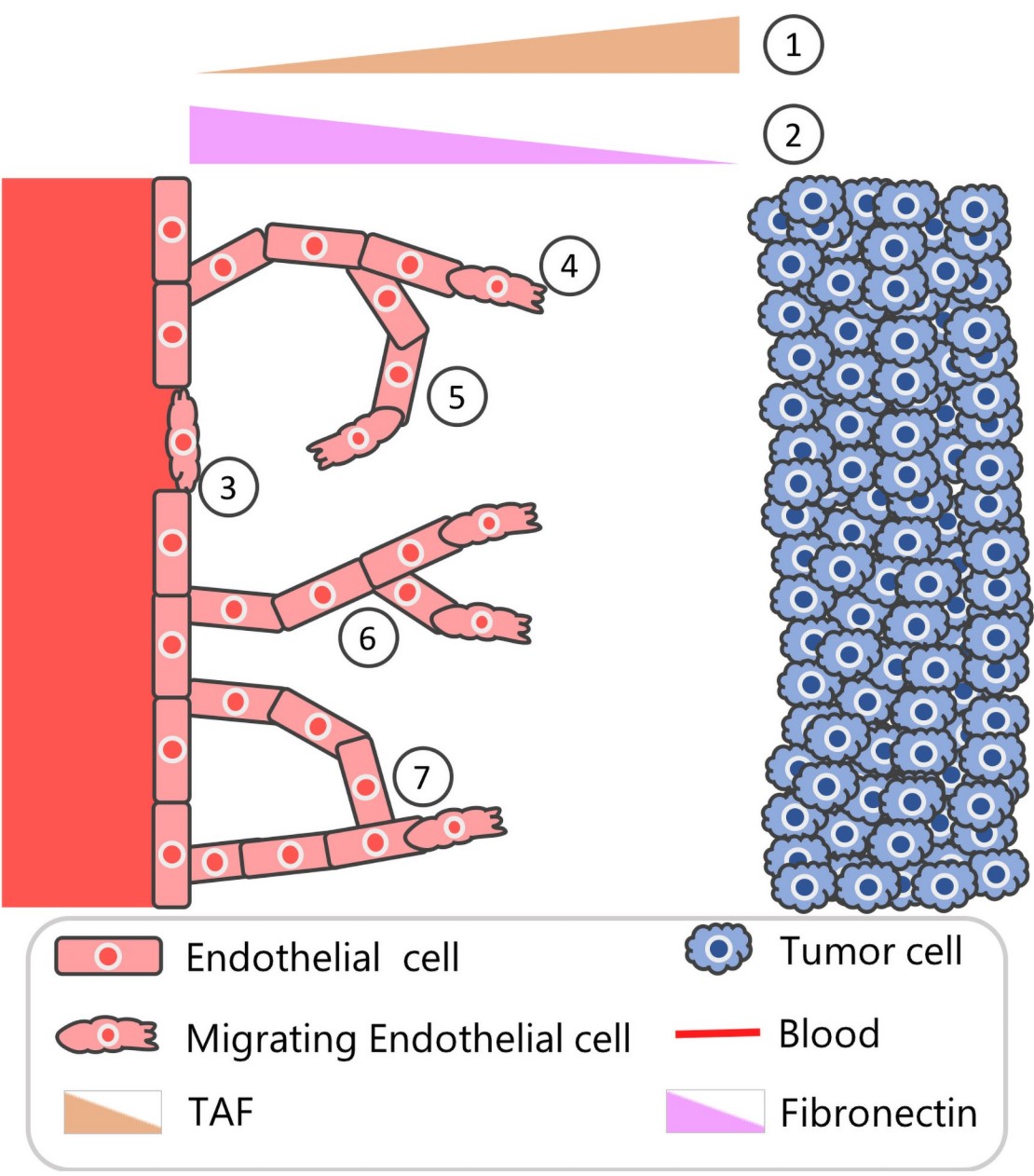

**Fig 1. Schematic overview of tumor-induced angiogenesis.** Schematic depicting seven aspects angiogenesis that are incorporated in the Anderson-Chaplain model of tumor-induced angiogenesis [17]. (1) Distant tumor cells release a range of chemoattractants, including vascular endothelial growth factors and basic fibroblast growth factors that stimulate the formation of new blood vessels. These growth factors are described collectively as a single, generic tumor angiogenesis factor (TAF). (2) Production and consumption of tissue-matrix bound fibronectin by the endothelial cells creates a spatial gradient of fibronectin across the domain. (3) Endothelial cells sense the TAF and fibronectin gradients and undergo individual cell migration. (4) As endothelial tip cells migrate, via chemotaxis, up spatial gradients of the TAF, stalk cells in the developing vessels are assumed to proliferate, creating what has been termed a "snail trail" of new endothelial cells. (5) Endothelial tip cells also migrate, via haptotaxis, up spatial gradients of fibronectin. (6) Endothelial cells in existing sprouts may initiate the formation of secondary sprouts. (7) If a sprout coincides with an existing vessel, then it is assumed to be annihilated and a new loop is formed; if two sprouts coincide or anastomose, then both are assumed to be annihilated and a new loop forms.

model proposed by Stokes and Lauffenberger [20]. It is an on-lattice model in which endothelial cells perform a biased random walk and generate blood vessel networks that resemble those observed experimentally [17]. While more detailed theoretical models have been developed (see reviews [21–24]), we focus on the Anderson-Chaplain model due to its simplicity and wide adoption in the mathematical biology literature. The methods presented in this work can also be used to study alternative models of angiogenesis, such as the phase-field model presented in [25], the 2D model of early angiogenesis and cell fate specification in [26], the 3D hybrid model presented in [27], and multiscale models of vascular tumor growth, such as those presented in [28–31].

The Anderson-Chaplain model describes the spatio-temporal evolution of three physical variables: endothelial tip cells, tumor angiogenesis factor (TAF), and fibronectin [17]. Fig 1 contains a model schematic where tumor cells located along the right boundary produce TAF and the endothelial tip cells from a nearby healthy vessel placed along the left boundary move via *chemotaxis* up spatial gradients of TAF. As the tip cells move, stalk cells located behind them proliferate, creating what has been termed a "snail trail" of new blood vessel segments [21]. Endothelial tip cells also migrate via *haptotaxis*, up spatial gradients of fibronectin, which is produced and consumed by the migrating endothelial cells and binds to the tissue matrix in which the cells are embedded. As the tip cells migrate and the stalk cells proliferate, the vessel sprouts elongate, and secondary sprouts may emerge from the vessels, at the end of a sprout or laterally from the nascent vessel. Further, when a tip cell collides with another vessel segment or another tip cell, the two cells may fuse, forming a closed loop or anastomosis.

We would like a quantitative method to understand and compare vessel networks that is applicable to experimental and synthetic data. Standard quantitative descriptors of vascular morphology proposed by [27, 32–35] include inter-vessel spacing, the number of branch points, measuring the total area (volume) covered for 2D (3D) simulations, vessel length, density, tortuosity, the number of vessel segments, and the number of endothelial sprouts. These metrics correlate with tumor progression and treatment response in experimental datasets [36]; however, they are computed at a fixed spatial scale, and neglect information about the connectedness (i.e., the topology) of vascular structures. Therefore, standard descriptors do not exploit the full information content of synthetic data generated from the models. At the same time, advances in imaging technology mean that it is now possible to visualize vascular architectures across multiple spatial scales [37]. These imaging developments are creating a need for new methods that can quantify the multi-scale patterns in vascular networks, and how they change over time.

A relatively new and expanding field of computational mathematics for studying the shape of data at multiple scales is TDA. TDA consists of algorithmic methods for identifying global structures in high-dimensional datasets that may be noisy [38, 39]. TDA has been useful for biomedical applications [40–43], including brain vessel MRI data [44] and experimental data of tumor blood vessel networks [45, 46]. A commonly-used tool from TDA is *persistent homology* (PH). *Homology* refers to the characterization of topological features (e.g., connected components and loops) in a dataset, and *persistence* refers to the extent to which these features persist within the data as a scale parameter varies. PH is a useful way to summarize the topological characteristics of spatial network data [45–47] as well as noisy and high dimensional agent-based models (ABMs) [48, 49]. More recently, PH has been combined with machine learning to extract accurate estimates of parameters from such models [50]. Most of these studies focus on PH calculations for point clouds of data; notably, PH has been adapted to analyze networks and grayscale images [51]. An active area of research is exploring different filtrations for application-driven PH.

In this paper, we develop a computational pipeline for analyzing simulations of the Anderson-Chaplain model over a range of biologically-relevant parameter values. Our main objectives are to identify model descriptor vectors that group together similar model simulations into *biologically interpretable* clusters. By biologically interpretable, we mean that simulations within a cluster arise from similar parameter values and simulations in different clusters arise from different parameter values.

We use two topological approaches, the sweeping plane and flooding filtrations, to construct simplicial complexes from binary image data generated from simulations of the Anderson-Chaplain model. We show that PH of the sweeping plane filtration and its subsequent vectorization provides an interpretable descriptor vector for the model parameters governing chemotaxis and haptotaxis. We show, by comparison with existing morphologically-motivated standard descriptor vectors, that this topological approach leads to more biologically interpretable clusters that stratify the haptotaxis-chemotaxis parameter space. Furthermore, the clusters generated from the sweeping plane filtration are robust and generalize well to unseen model simulations.

## Materials and methods

We simulate the Anderson-Chaplain model of tumor-induced angiogenesis for a range of values of the haptotaxis and chemotaxis coefficients, $\rho$ and $\chi$. Each simulation generates vasculature composed of a collection of interconnected blood vessels in the form of binary image data. We compute three so-called standard descriptor vectors to summarize each simulation: two are computed at 50 time steps and the third is computed at the final time step. We also compute 30 topological descriptor vectors. These topological descriptors encode the proposed multiscale quantification of the vascular morphology. We calculate standard and topological descriptor vectors for the entire set of 1,210 simulations. We then analyze the simulations by clustering the standard or topological descriptor vectors and compare the results. Fig 2 shows the pipeline of data generation and analysis. We give details of the Anderson-Chaplain model, including governing equations, parameter values and implementation, in the S1 Appendix. The Python files and notebooks used for all steps of our analysis are publicly available at https://github.com/johnnardini/Angio_TDA.

### Data generation

We simulate the Anderson-Chaplain model as the parameters $\rho$ and $\chi$ are each independently varied among the following 11 values: {0, 0.05, 0.1, 0.15, . . ., 0.5}. The parameter bounds of [0, 0.5] were chosen to yield a range of vascular architectures that are visibly distinct and recapitulate vascular patterns seen *in vivo*. The resolution of the parameter mesh (11 discrete values between 0 and 0.5) was chosen for computational tractability while still being sufficient to illustrate how TDA methods can detect and quantify differences between simulations from similar parameter values that are difficult to distinguish visually. We generate 10 realizations of the model for each of the 121 $(\rho, \chi)$ parameter combinations, and produce 1,210 binary images that summarize the different synthetic blood vessel networks. Each simulation is initialized using the same initial conditions, with all other model parameters fixed at the baseline values used in [17], and no-flux conditions imposed on the domain boundary. For consistency with [17], all simulations are computed on lattices of size $201 \times 201$. Each simulation continues until either an endothelial tip cell crosses the line $x = 0.95$ (in dimensionless spatial units) or the maximum simulation time, $t = 20$ (in dimensionless time units), is exceeded. The final model output is a $201 \times 201$ binary image in which nonzero (zero) pixels denote the presence (absence) of an endothelial cell at that lattice site.

## 1. Spatio-temporal modeling: Anderson-Chaplain model

**Fig 2. Data generation and analysis pipeline.** (1) <u>Spatio-temporal modeling: Anderson-Chaplain model.</u> The Anderson-Chaplain model is simulated for $11 \times 11 = 121$ different values of the haptotaxis and chemotaxis parameters, $(\rho, \chi)$. The model output is saved as a binary image, where pixels labeled 1 or 0 denote the presence or absence, respectively, of blood vessels. We generate 10 realizations for each of the 121 parameter combinations, leading to 1,210 images of simulated vessel networks. (2) <u>Data analysis.</u> We use the binary images from Step 1 to generate standard and topological descriptor vectors A.) Standard descriptor vectors We compute the number of active tip cells and the number of vessel segments at discrete time points. We also compute the length of the vessels at the final time point [32, 34, 35, 52]. B.) Topological descriptor vectors. We construct flooding and sweeping plane filtrations [44] using binary images from the final timepoint of each simulation. We track the birth and death of topological features (connected components and loops) that are summarized as Betti curves and persistence images [52]. (3) <u>Data clustering.</u> We perform $k$-means clustering using either the standard or topological descriptor vectors computed during step 2 from all 1,210 simulations. In this way, we decompose $(\rho, \chi)$ parameter space into regions that group vessel networks with similar morphologies.

**Morphologically-motivated standard descriptors and descriptor vectors.** Morphologically-motivated standard descriptors are used to summarize physical characteristics of each model simulation. Commonly used descriptors include inter-vessel spacing, the number of branch points, the total area (volume) spanned by the network for 2D (3D) simulations, the

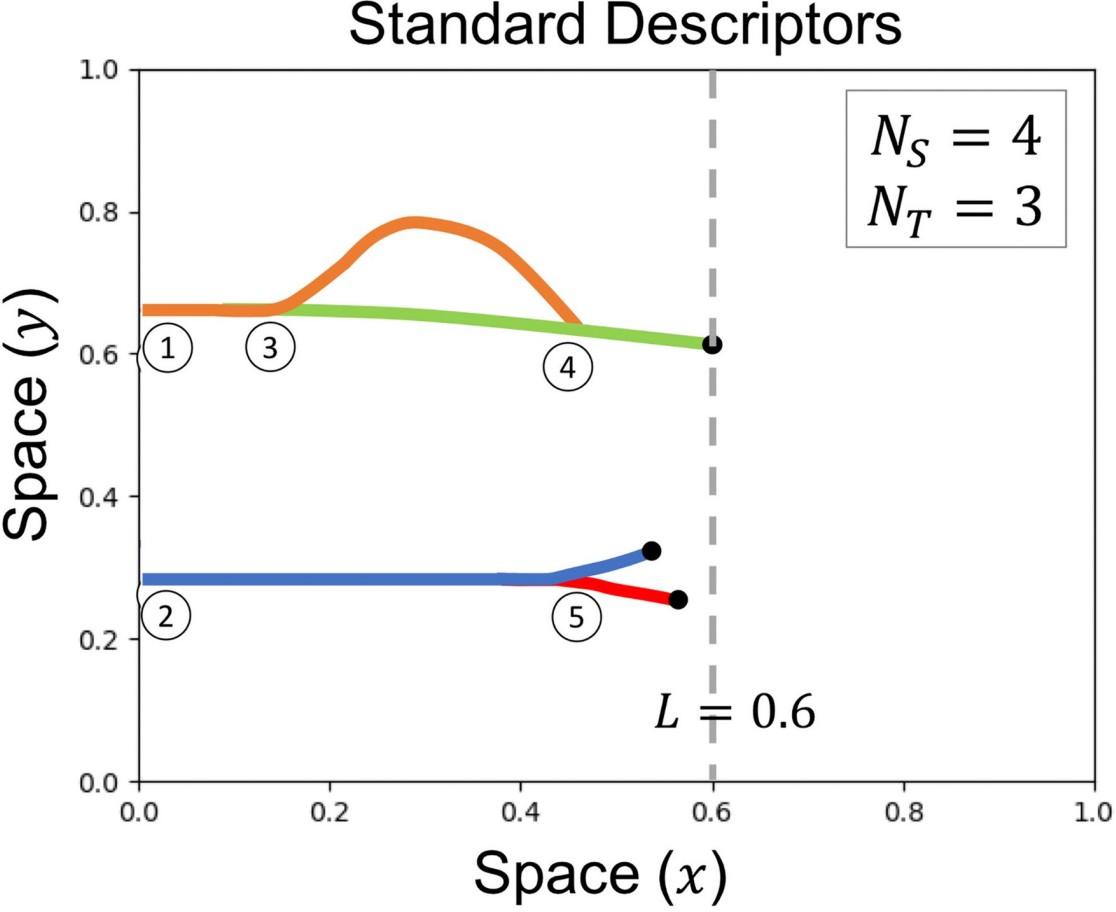

**Fig 3. Standard descriptor overview.** Standard descriptors are computed for each model simulation include the number of vessel segments ($N_S$, depicted by each colored solid curve), the number of endothelial tip cells ($N_T$, depicted by black dots), and the $x$-coordinate of the vessel segment location with the greatest horizontal distance from the $y$-axis ($L$, depicted by the vertical grayed dashed line). This schematic example is initialized with 2 vessel segments and 2 tip cells at locations (1) and (2). A branching event at location (3) increases the value of both $N_S$ and $N_T$ by one. An anastomosis event at location (4) decreases $N_T$ by one. The branching event at location (5) increases both $N_S$ and $N_T$ by one.

shape of such loops in 2D (voids in 3D), and tortuosity [27, 32–35, 53]. We choose to use the following standard descriptors due to their prevalence in the literature: the number of vessel segments ($N_S$), the number of endothelial tip cells ($N_T$), and the length of the vessel that has penetrated furthest from the parent vessel ($L$) [18, 32, 34, 35]. In Fig 3 we use idealized networks to indicate how the standard descriptors are defined. As described in S1 Appendix, $N_S$ and $N_T$ both increase by one with each branching event, and $N_T$ decreases by one with each anastomosis event. We record $N_S$ and $N_T$ at 50 time steps to create the descriptor vectors. To facilitate downstream clustering analysis, we require vectors of the same length for all simulations. To ensure the dynamic descriptor vectors for each model simulation are of length 50, we record $N_S(\mathbf{t})$ and $N_T(\mathbf{t})$ at the 50 time steps $t = \{t_i\}_{i=1}^{50}$, where $t_i = (i-1)\Delta t$, $\Delta t = t_{\text{end}}/49$, and $t_{\text{end}}$ denotes the duration of a given simulation. We remark that the requirement for having vectors of the same length for all simulations might not be feasible when dealing with experimental data, which could include heterogeneities in sample size and time point locations. We note further that this requirement only applies to the standard descriptors; for the topological

filtrations considered in future sections, only the final segmented vessel image is needed. For each simulation, we also compute $L(t_{\text{end}})$, which we define as the $x$-coordinate of the vessel segment location which has the greatest horizontal distance from the $y$-axis. Thus, the standard descriptor vectors we use for subsequent clustering analysis are $N_S(\mathbf{t})$, $N_T(\mathbf{t})$ and $L(t_{\text{end}})$.

## Quantifying vessel shape using topological data analysis

The most prominent method from TDA is persistent homology (PH) [38, 51, 54]. PH is rooted in the mathematical concept of homology, which captures the characteristics of shapes. To compute (persistent) homology from data, one needs to first construct simplicial complexes, which can be thought of as collections of generalized triangles. From the constructed simplicial complexes, one can quantify and visualize the connected components (dimension 0) and loops (dimension 1) of a dataset at different spatial scales. While we compute the $N_T$ and $N_S$ standard descriptor vectors at 50 time points for each model simulation (see Section **Morphologically-motivated standard descriptors and descriptor vectors**), we compute topological descriptor vectors from the final output binary image.

**Simplicial complexes and homology.** We construct simplicial complexes from data points derived from binary images of the completed simulations of the Anderson-Chaplain model (see "Synthetic data" in Step 1 of Fig 2, where red pixels denote presence of vasculature). Normally, TDA of image data lends itself to the construction of cubical complexes, which are generalizations of cubes; however, we have binary rather than grayscale data, so we instead use information about the model generation and Moore neighborhoods as we describe here. Each pixel that has a value of one is embedded in $\mathbb{R}^2$ at the centroid of the pixel as a vertex (or 0-simplex), as demonstrated in Fig 4. We term the resulting collection of points a *point cloud*. We connect two points by an edge (or 1-simplex) if they are within each others' Moore neighborhood (the Moore neighborhood for one pixel is shaded in Fig 4B). If three points are pairwise connected by an edge, then we connect them with a filled-in triangle (or 2-simplex). The union of 0-, 1-, and 2-simplices form a *simplicial complex*, as shown in Fig 4C.

To compute topological invariants, such as connected components (dimension 0) and loops (dimension 1), we use homology. To obtain homology from a simplicial complex, $K$, we construct vector spaces whose bases are the 0-simplices, 1-simplices, and 2-simplices, respectively, of $K$. There is a linear map between 2-simplices and 1-simplices, called the boundary

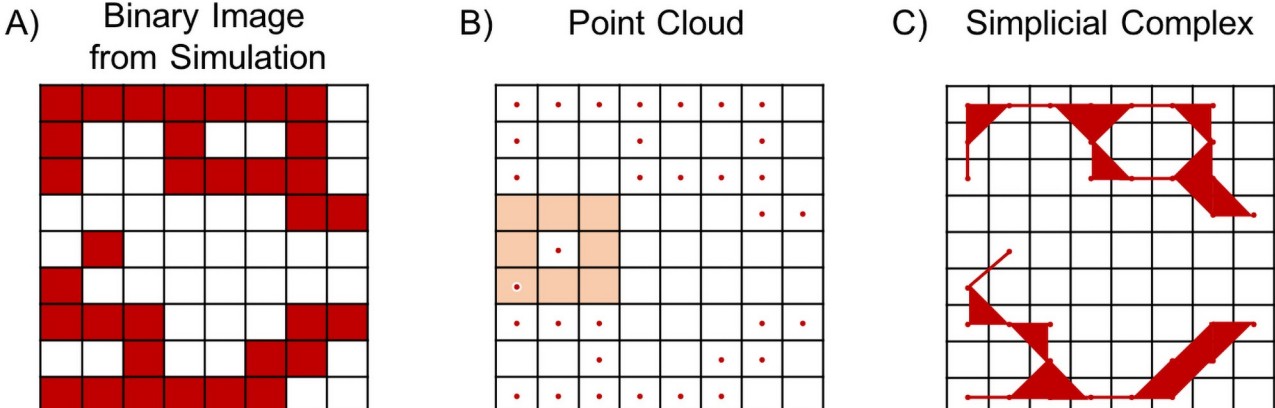

**Fig 4. Converting a binary image into a simplicial complex.** A) Schematic binary image. B) Point cloud for all nonzero pixels in the binary image. The Moore neighborhood of one nonzero pixel is highlighted in salmon. C) A simplicial complex constructed from the point cloud in panel B.

map $\partial_2$, which sends triangles to their edges. Similarly, the boundary map $\partial_1$ sends edges to their end points and $\partial_0$ sends points to 0. The action of the boundary map $\partial_1$ on the simplices is stored in a binary matrix where the entry $a_{i,j}$ indicates whether the $i$-th 0-simplex forms part of the boundary of the $j$-th 1-simplex. If so, then $a_{i,j} = 1$; otherwise, $a_{i,j} = 0$. One can compute the kernel Ker($\cdot$) and image Im($\cdot$) of the boundary maps to obtain the vector spaces $H_0(K) =$ Ker($\partial_0$)/Im($\partial_1$) and $H_1(K) =$ Ker($\partial_1$)/Im($\partial_2$). These vector spaces are also referred to as homology groups and their dimensions define topological invariants called the *Betti numbers* of $K$, $\beta_0$ and $\beta_1$, which quantify the numbers of connected components and loops, respectively.

**Filtrations for simulated vasculature.** There are different ways to study vascular data at multiple scales [44–46, 55]. We encode the multiple spatial scales of the data using a *filtration*, which is a sequence of embedded simplicial complexes $K_0 \subseteq K_1 \subseteq \ldots \subseteq K_{\text{end}}$ built from the data. A challenge for researchers is to determine informative filtrations for specific applications [56], such as blood vessel development in our study. Here, the input data is the binary image from the final timepoint of the Anderson-Chaplain model, which we call $N$ (See Eq (13) in S1 Appendix). We construct sequences of binary images that correspond to different filtered simplicial complexes: a sweeping plane filtration [44] and a flooding filtration [55]. We remark that both filtrations can be considered sublevelset filtrations corresponding to a height function $h : X \to \mathbb{R}$ on some simplicial complex $X$ (or just on the vertices/pixels and considering the simplicial subcomplex spanned by them).

**Sweeping plane filtration.** In the sweeping plane filtration, we move a line across the binary image $N$ and include pixels in the filtration at discrete steps once a pixel is crossed by the line. On every filtration step, we move the line by a fixed number of pixels in a chosen direction. In Fig 5, we illustrate the method when a vertical line moves from left to right (LTR). From the corresponding sequence of binary images, we construct a filtered simplicial complex $K^{\text{LTR}} = \{K_1, K_2, \ldots, K_{\text{end}}\}$. Similarly, we construct the filtered simplicial complexes $K^{\text{RTL}}$ when the vertical line moves from right to left (RTL), $K^{\text{TTB}}$ when a horizontal line moves from top to bottom (TTB), and $K^{\text{BTT}}$ when a horizontal line moves from bottom to top (BTT).

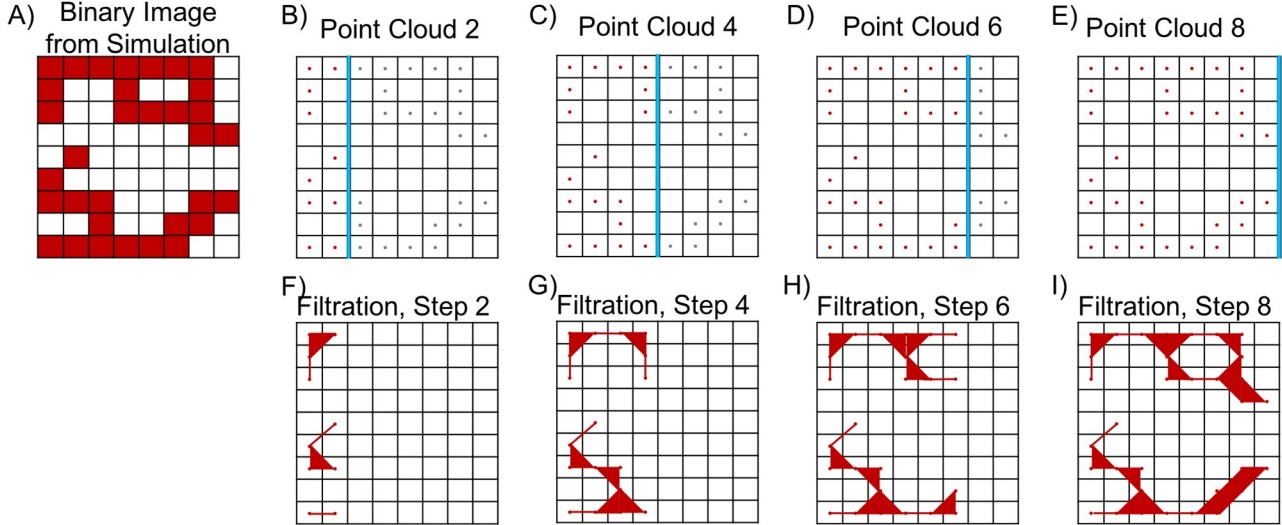

**Fig 5. The LTR sweeping plane filtration for a binary image.** A) Sample binary image from a simulation of the Anderson-Chaplain model. B-E) Point clouds used for each iteration of the sweeping plane approach. On each step, only pixels located to the left of the plane (denoted here with a blue line) are included; gray pixels are ignored. F-I) Filtration steps constructed from the point clouds in panels B-E.

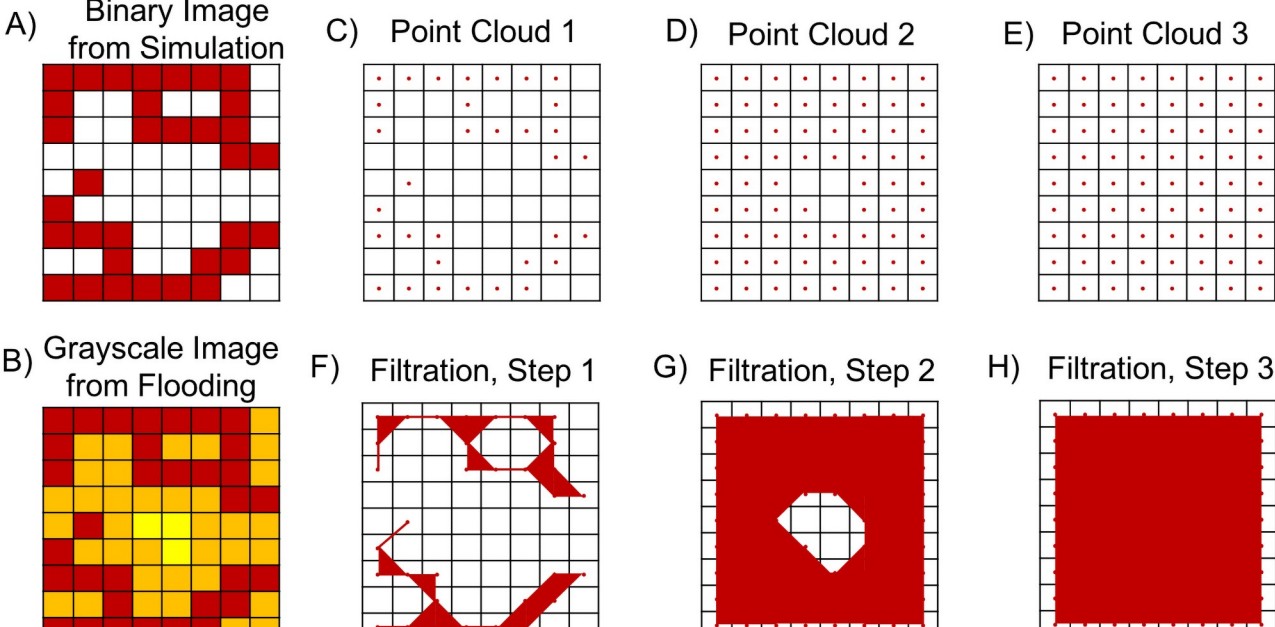

**Fig 6. The flooding filtration for a binary image.** A) Schematic binary image. B) Grayscale image of the original binary image. Red pixels are nonzero in the initial image, orange pixels become nonzero after one round of flooding, and yellow pixels become nonzero after two rounds of flooding. C-E) We performed three filtrations by dilating the image twice. During each filtration, the eight neighboring pixels of all nonzero pixels (ie pixels not marked as white) become nonzero. Red pixels are nonzero in the original image, orange pixels become nonzero on the second filtration (i.e., after one step of flooding), and yellow pixels become nonzero on the third filtration (i.e., after two steps of flooding). F-H) Simplicial complexes associated with the filtrations presented in panels C-E.

**Flooding filtration.** Our second filtration is the flooding filtration. From the binary image, $N$, associated with a model simulation, we construct a sequence of binary images in the following way. On the first step, we input the output binary image from a model simulation. In the binary image, pixels with value one denote the presence of an endothelial cell and pixels with value zero denote the absence of endothelial cells. To create the second binary image, we consider all pixels of value one and assign all pixels in their Moore neighborhood (as shown in Fig 4B) to value one. We repeat this process until a binary image with only pixels of value one is created. In Fig 6B, for example, the red pixels denote the nonzero pixels from the initial binary image, the orange pixels denote the nonzero pixels added on the first round of flooding, and the yellow pixels denote the nonzero pixels added on the second round of flooding. Each "round" is called a step. From the corresponding sequence of binary images, we construct a filtered simplicial complex $K^{\text{flood}} = \{K_1, K_2, \ldots, K_{\text{end}}\}$. We choose $K_{\text{end}} = 25$ so that the Betti curves for all simulations attain $\beta_0 = 1$ and $\beta_1 = 0$ at the final filtration step, thus ensuring that the maximum amount of topological information is encoded in the Betti curves.

**Persistent homology (PH).** PH is an algorithm that takes in data via a filtration and outputs a quantification of topological features such as connected components (dimension 0) and loops (dimension 1) across the filtration. The simplicial complexes are indexed by the scale parameter of the filtration. In our case, the scale parameter corresponds to the spatial location in the sweeping plane filtration or steps of flooding in the flooding filtration. Note that we compute these filtrations on simulated blood vessel network images at the final time step of a model simulation. The inclusion of a simplicial complex $K_i \subseteq K_j$ for $i \leq j$ gives a relationship between the corresponding homology groups $H_p(K_i)$ and $H_p(K_j)$ for $p = 0, 1$. This relationship enables us to track topological features such as loops along the simplicial complexes in the

filtration. Intuitively, a topological feature is born in filtration step $b$ when it is first computed as part of the homology group $H_p(K_b)$ and dies in filtration step $d$ when that feature no longer exists in $H_p(K_d)$, i.e., when a connected component merges with another component or when a loop is covered by 2-simplices. The output from PH is a multiset of intervals $[b, d)$ which quantifies the persistence of topological features. Each topological feature is said to persist for the scale $d - b$ in the filtration. Our topological descriptors are the connected components (i.e., 0-dimensional) and loops (i.e., 1-dimensional features) obtained by computing persistent homology of the sweeping plane and flooding filtrations. We use the superscript notation $K^v$ to denote the filtered simplicial complex $K$ in direction $v = \{$LTR, RTL, TTB, BTT, flood$\}$.

**Topological descriptor vectors from PH.** We use PH to generate different output vectors.

- **Betti curves.** Betti curves [57] show the Betti numbers on each filtration step. Let $\beta_i(K^v)$ denote the Betti curves of filtered simplicial complex $K$ in direction $v$ for dimensions $i = \{0, 1\}$.

- **Persistence images.** A persistence image [52] uses as input the birth-death pairs given by PH and converts the set of (birth $b$, persistence $(d - b)$) coordinates into a vector, a format which is suitable for machine learning and other classification tasks. Following the standard definition of a persistence image, the coordinates $(b, d - b)$ are blurred by a Gaussian, with standard deviation $\sigma$, that is centered about each birth-persistence point, which accounts for uncertainty [52]. We take a weighted sum of the Gaussians and place a grid on this surface to create a vector. We compute two alternative weighting strategies: all ones (in which case all features are equally weighted) and a persistence weight of $(d - b)$ (in which more persistent features are given larger weights). Let $\text{PIO}_i(K^v)$ and $\text{PIR}_i(K^v)$ denote the corresponding persistence images computed from $K^v$ with equal and ramped weighting, respectively, for dimension $i = \{0, 1\}$.

The topological descriptor vectors that we compute for each model simulation are $\beta_i(K^v)$, $\text{PIO}_i(K^v)$, and $\text{PIR}_i(K^v)$, where $i = \{0, 1\}$ and $v = \{$LTR, RTL, TTB, BTT, flood$\}$. In total each simulation has $3 \times 2 \times 5 = 30$ topological descriptor vectors.

## Simulation clustering

We use an unsupervised algorithm, $k$-means clustering, to group the quantitative descriptor vectors of the synthetic data generated from simulations of the Anderson-Chaplain model. To cluster $n$ samples $x_1, x_2, \ldots, x_n \in \mathbb{R}^d$ into $k$ clusters, the $k$-means algorithm finds $k$ points in $\mathbb{R}^d$, clusters each sample according to which of the $k$ means it is closest to, and then minimizes the within-cluster residual sum of squares distance [58]. We cluster the standard descriptor vectors and the topological descriptor vectors (see steps 2A and 2B of the methods pipeline in Fig 2), respectively, to investigate whether descriptor vectors lead to biologically interpretable clusters.

We test the robustness of each clustering assignment by randomly splitting the descriptor vectors into training and testing sets, and use the testing set to evaluate out-of-sample (OOS) accuracy. Specifically, 7 of the 10 descriptor vectors from each of the 121 $(\rho, \chi)$ parameter combinations are randomly chosen with uniform sampling and placed into a training set (847 simulations); the remaining 3 descriptor vectors from each $(\rho, \chi)$ combination are placed into a testing set (363 simulations). After performing unsupervised clustering with a $k$-means model on the training set, each $(\rho, \chi)$ combination is labeled by the majority of cluster assignments among its 7 training simulation descriptor vectors. To evaluate each clustering model's OOS accuracy, the 3 test data samples for each $(\rho, \chi)$ parameter combination are given a

"ground-truth" label equal to the cluster assignment for the 7 training simulations for that same $(\rho, \chi)$ parameter combination. A "predicted" label is calculated for each descriptor vector from the test data using the trained $k$-means model, and OOS accuracy is defined as the proportion of test simulations for which the "predicted" label matches the "ground-truth" label. For example, if the seven training simulations from the parameter combination $(\rho, \chi) = (0.2, 0.2)$ are placed into clusters $\{1, 1, 2, 2, 1, 1, 1\}$, then the three testing simulations from $(\rho, \chi) = (0.2, 0.2)$ will be labeled as being in cluster 1. If the test simulations are predicted to be from clusters $\{1, 2, 1\}$, then we compute an OOS accuracy of 66.67%. We use all 363 testing simulations when computing OOS accuracy and use the `KMeans` command from scikit-learn (version 0.22.1), which uses Lloyd's algorithm for Euclidean space, for training and prediction.

## Results

We simulated the Anderson-Chaplain model for 121 different values of the chemotaxis and haptotaxis coefficients $(\rho, \chi)$ and performed 10 model realizations for each parameter pair. We observed that simulations generated by different parameter pairs may appear visually distinct in their vessel growth patterns. To quantify and summarize the morphologies, we computed dynamic standard descriptor vectors as well as topological descriptor vectors for the set of 1,210 model simulations. We then performed unsupervised clustering using either standard or topological descriptor vectors to partition the $(\rho, \chi)$ parameter space.

As we detail here, we found that the topological sweeping plane descriptor vectors offered a multi-scale analysis that led to biologically interpretable and robust clusterings of the $(\rho, \chi)$ parameter space, whereas the clusterings of the standard descriptor vectors were not biologically interpretable. The topological analysis succeeded in discriminating fine-grained vascular structures and the parameter pairs that generated them.

### Haptotaxis and chemotaxis alter vessel morphology

We investigated the influence of haptotaxis and chemotaxis by varying the parameters $\rho$ and $\chi$ We note that similar parameter sensitivity analyses have been performed previously [17, 22]; we reproduce this analysis as the basis for our data analysis. Recall that in model simulations, the vessels emerge from the left-most boundary and are recruited towards a tumor located on the right boundary. When chemotaxis alone drives endothelial tip cell movement ($\rho = 0.00$, $\chi = 0.50$), the blood vessel segments are long and thin as the tip cells migrate directly towards the tumor and produce individual vessel segments with few anastomosis events (Fig 7A). When haptotaxis drives tip cell movement ($\rho = 0.50$, $\chi = 0.00$), the vessels fail to reach the tumor (Fig 7B). When both haptotaxis and chemotaxis are active ($\rho = \chi = 0.15$), the network spans a large portion of the spatial domain and is highly connected (Fig 7C).

### Standard and topological descriptors measure distinct aspects of vascular architectures

Rather than performing visual inspection of all model simulations, we computed and compared quantitative vessel descriptor vectors using data science approaches as described in the **Materials and Methods**. We demonstrate here how the biological and topological descriptors are distinct from each other. In Fig 8, we observe how the number of tip cells ($N_T$), vessel segments ($N_S$), connected components ($\beta_0$), and loops ($\beta_1$) associated with an evolving vasculature change following different events. After each anastomosis event, $N_T$ decreases by one (Fig 8B). After a branching event, both $N_T$ and $N_S$ increase by one (Fig 8C). We conclude that $N_T$ is unchanged and $N_S$ increases by one after one branching event and one anastomosis event have occurred. By contrast, the values of the topological descriptors before and after these events are

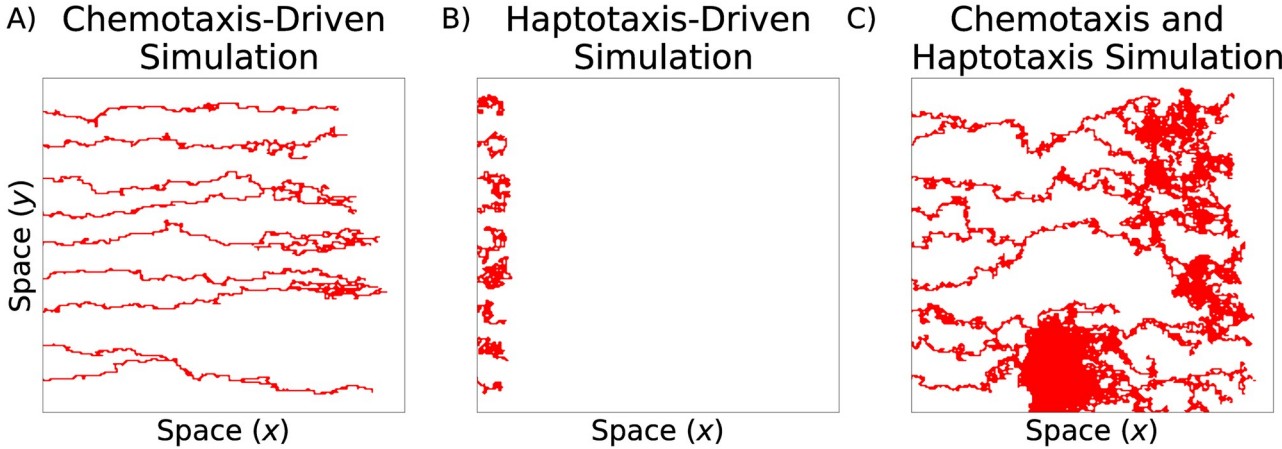

**Fig 7. Typical realizations of the Anderson-Chaplain model.** A) Chemotaxis-driven tip cell movement ($\rho = 0.00$, $\chi = 0.50$), B) haptotaxis-driven tip cell movemement ($\rho = 0.50$, $\chi = 0.00$), and C) tip cell movement driven by chemotaxis and haptotaxis ($\rho = 0.15$, $\chi = 0.15$).

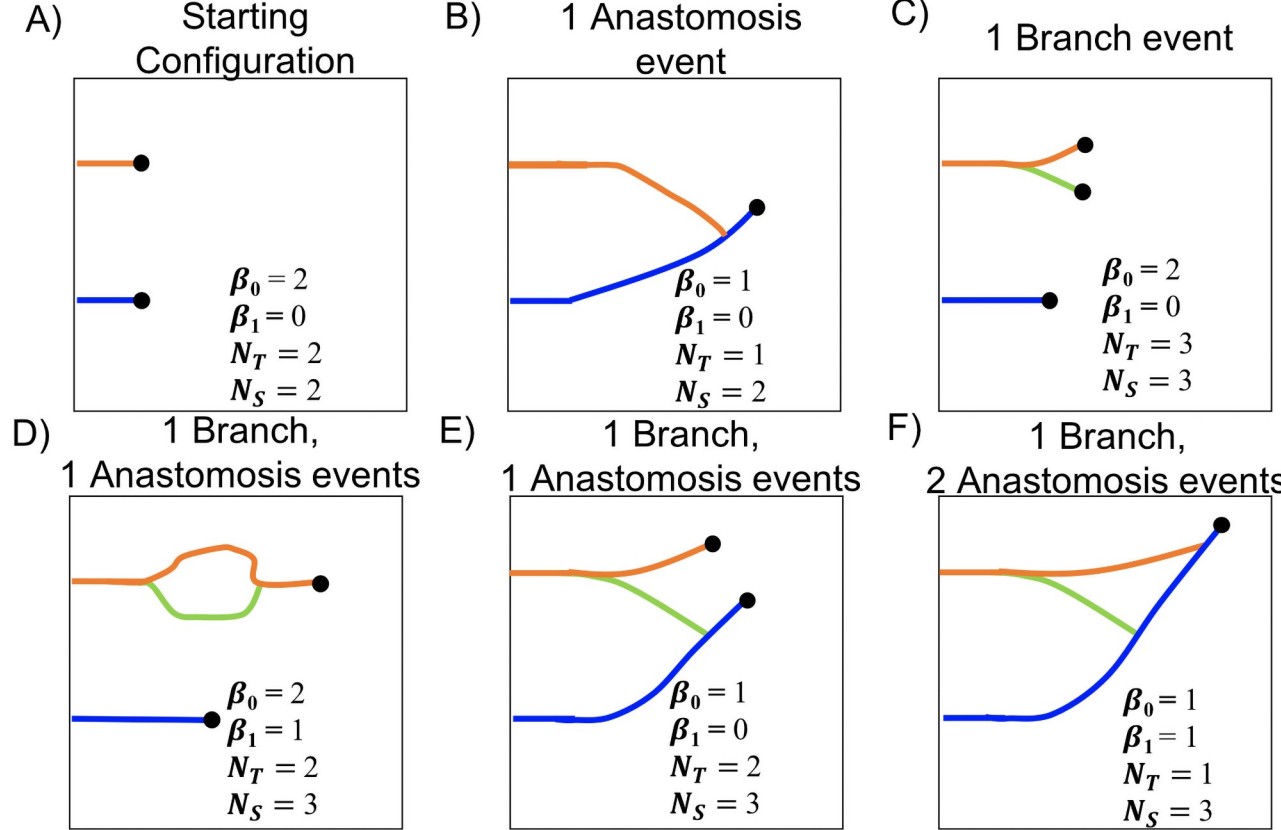

**Fig 8. Standard and topological descriptors describe distinct aspects of vessel morphology.** For clarity, each vessel segment is colored differently, and black dots represent active tip cells. A) The initial vessel configuration. B) A schematic showing how the vascular architecture changes after an anastomosis event. C) A schematic showing how the vascular architecture changes after a branching event. D-E) Schematics showing two ways in which vasculature can change after one anastomosis event and one branching event. F) A schematic showing how the vessels can change after one branching and two anastomosis events. (Key: $\beta_0$, number of connected components; $\beta_1$, number of loops; $N_T$, number of tip cells; $N_S$, number of vessel segments).

not necessarily identical, as depicted in Fig 8D and 8E. When an anastomosis event involves vessel segments that were previously connected, the number of loops, $\beta_1$, increases by 1 (compare Fig 8A and 8D); when the vessel segments were not previously connected, $\beta_0$ decreases by 1 (compare Fig 8A–8E).

Let $N_A(r, s)$ and $N_B(r, s)$ denote the numbers of anastomosis and branching events, respectively, that occur between times $t = r$ and $t = s$. We computed $N_T(t)$, $N_S(t)$ vectors of the standard descriptors of evolving blood vessel simulations, as follows:

$$
\begin{aligned}
N_T(t_1) &= N_S(t_1) = 9, \\
N_T(t_i) &= N_T(t_{i-1}) + N_B(t_{i-1}, t_i) - N_A(t_{i-1}, t_i), \ \ i = 2, ..., 50, \\
N_S(t_i) &= N_S(t_i) + N_B(t_{i-1}, t_i), \ \ i = 2, ..., 50.
\end{aligned}
$$

The values of $N_T(t_1)$ and $N_S(t_1)$ were determined from the models' initial conditions.

We first constructed the filtrations $K^\nu$ where $\nu$ = {LTR, RTL, TTB, BTT, flood} of each simulation's final binary image. Then we computed connected components and loops, which we call topological descriptors of $K^\nu$, using PH. The topological descriptor vectors, Betti curves and persistence images, were then constructed for these five filtrations.

## Quantification of blood vessel architecture data

Fig 9 shows the standard descriptor vectors for the three simulations presented in Fig 7. For the haptotaxis-driven simulation growth in the horizontal direction halts (the total length remains at 0.095 units) and the numbers of tip cells and vessel segments stay constant (at or near the initial condition of 9 vessels). By contrast, the number of tip cells and vessel segments increase over time for the chemotaxis-driven simulation, with a more marked increase when both chemotaxis and haptotaxis are active. For both chemotaxis-active simulations, the vessels extend to the maximum horizontal distance.

We next illustrate how we use the plane sweeping and flooding filtrations to analyze the chemotaxis-driven, haptotaxis-driven, and chemotaxis and haptotaxis-driven simulations (from Fig 7). We computed Betti curves, $\beta_0(K^\nu)$ and $\beta_1(K^\nu)$, of the simulated data with sweeping plane filtrations for $\nu$ = {LTR, RTL, TTB, BTT} (see Fig 10) and a flooding filtration $\nu$ = {flood} (see Fig 11). These filtrations are constructed to provide complementary but not directly comparable information; roughly, the sweeping plane gives an indication of the

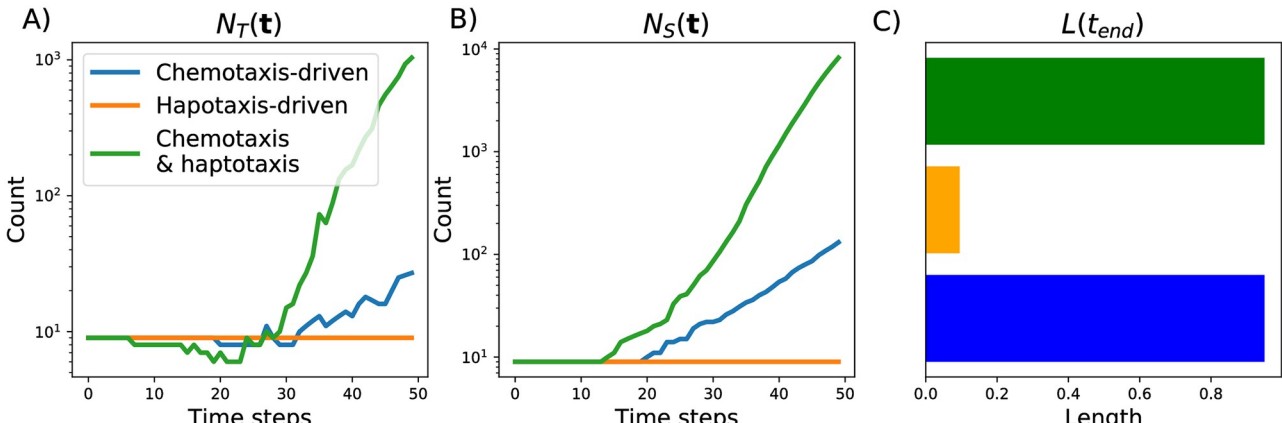

**Fig 9. The standard descriptors used to summarize model simulations.** We show how the following descriptors change over 50 time steps for the three simulations presented in Fig 7: A) the number of tip cells, $N_T(\mathbf{t})$, B) the number of vessel segments, $N_S(\mathbf{t})$, and C) the final vessel length, $L(t_{end})$.

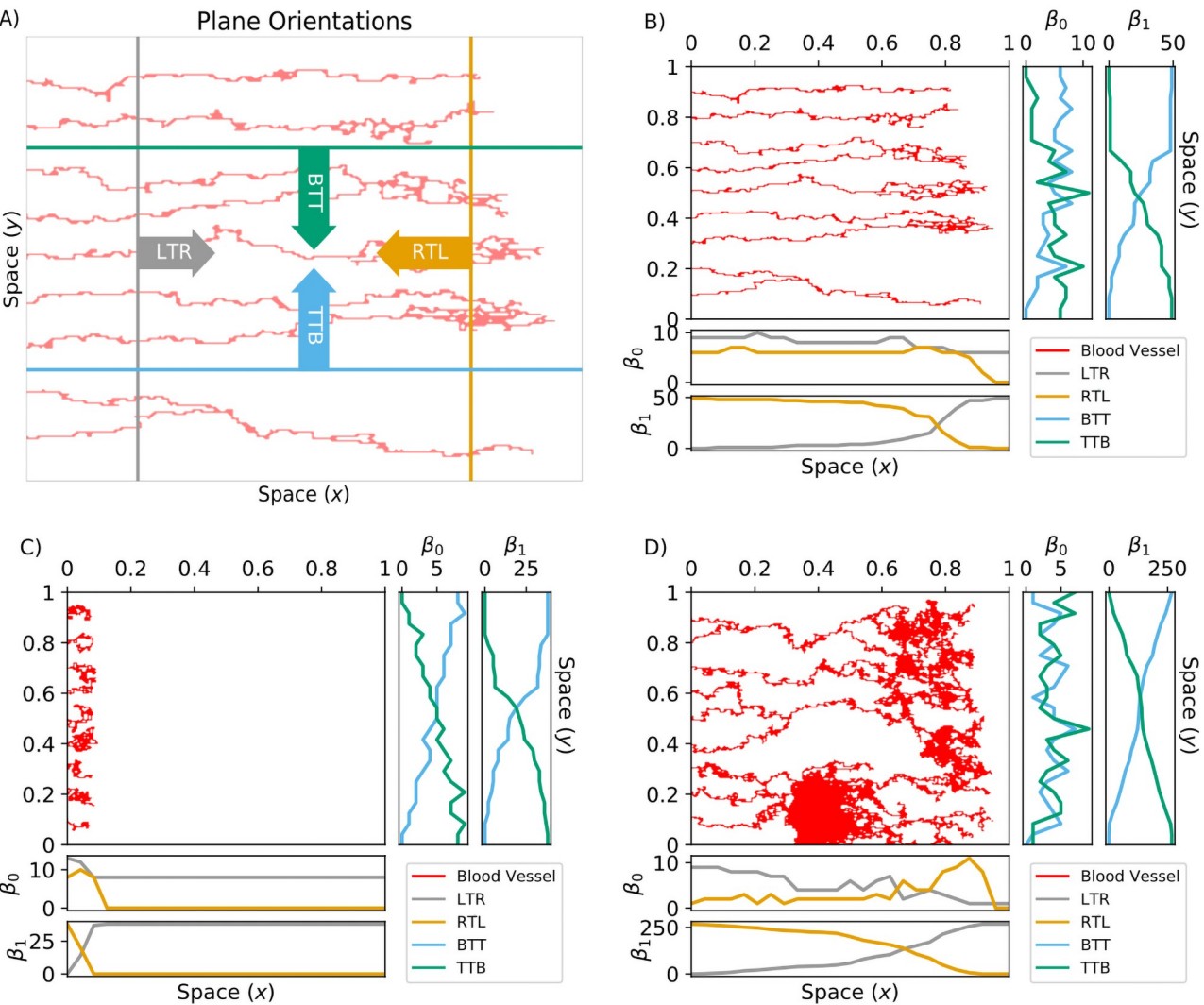

**Fig 10. TDA sweeping plane descriptor vectors.** A) The sweeping plane directions are left-to-right (LTR, gray); right-to-left (RTL, orange); bottom-to-top (BTT, cyan); and top-to-bottom (TTB, green). For these four $v = iTj$ filtrations, we started with points on the $i$th boundary and included more points as we stepped towards the $j$th boundary. Repeating this process produced the spatial filtration $K^v = K_0, K_1 \ldots K_{\text{end}}$ for $v = \{\text{LTR}, \text{RTL}, \text{TTB}, \text{BTT}\}$. B-D) We formed the Betti curves $\beta_0(K^v)$ and $\beta_1(K^v)$ by computing the Betti numbers along each step of $K^v$ for the three blood vessel simulations presented in Fig 7; B) the chemotaxis-driven simulation, C) the haptotaxis-driven simulation and D) chemotaxis and haptotaxis simulation.

network in the (x-y) plane whereas the flooding filtration also focusses on vessel density (see earlier description of filtration construction). We could interpret the PH of the sweeping plane filtration direction as follows: the blood vessels grow primarily from left to right, so the LTR and RTL filtrations primarily identify dynamic changes in branching and anastomosis of vessels. The BTT and TTB filtrations are similar to each other, and reflect only horizontal changes, which could be interpreted as measuring bendiness or tortuosity. Therefore, we only discuss the BTT results in this section.

For the chemotaxis-driven simulation (Figs 10B, 11A and 11B), the $\beta_0(K^{\text{LTR}})$ descriptor vector slightly decreases with distance $x$ from the $y$-axis as vessel segments anastomose together. The $\beta_0(K^{\text{RTL}})$ and $\beta_1(K^{\text{RTL}})$ descriptor vectors decrease with $x$ at the end of each vessel segment. The magnitude of $\beta_0(K^{\text{BTT}})$ increases almost linearly with the $y$-spatial coordinate, and $\beta_1(K^{\text{BTT}})$ increases with $y$ until it plateaus for $y \geq 0.6$. The $\beta_0(K^{\text{flood}})$ descriptor vector decreases

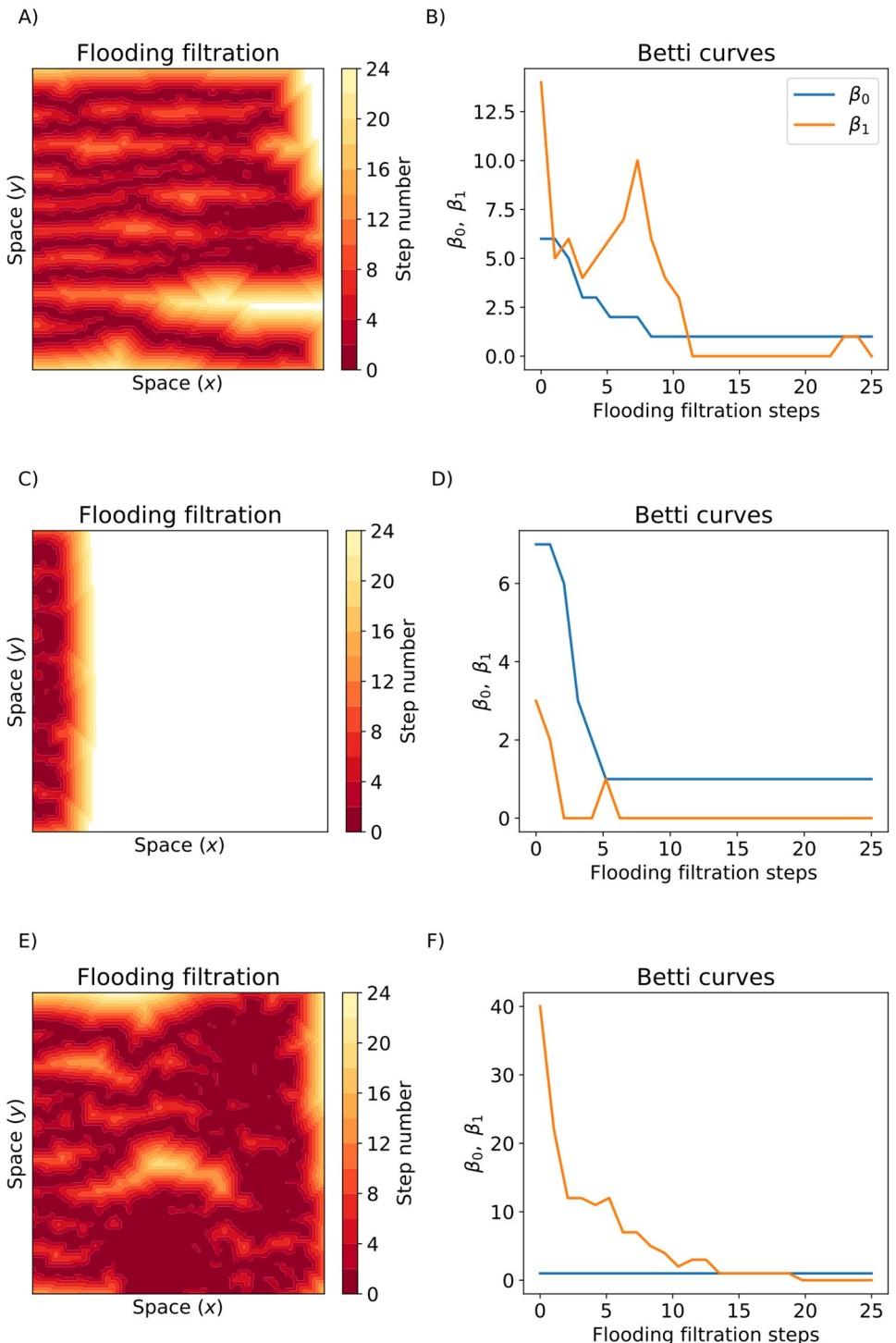

**Fig 11. The flooding filtration.** A,C,E) Illustrate the flooding filtrations for the three blood vessel simulations from Fig 7. Dark red pixels were points included in the first steps of the filtration and yellow pixels were the points included in later filtration steps. White pixels were not included after 24 steps of the flooding filtration. B,D,F) The Betti Curves ($\beta_0(K^{\text{flood}})$ and $\beta_1(K^{\text{flood}})$) for the chemotaxis-driven, haptotaxis-driven, and chemotaxis and haptotaxis simulations respectively.

with each filtration step, and $\beta_1(K^{\text{flood}})$ begins with $\beta_1 = 14$ and achieves a local maximum of $\beta_1 = 10$ after seven filtration steps before reaching $\beta_1 = 0$ after 11 filtration steps.

For the haptotaxis-driven simulation (Figs 10C, 11C and 11D), all LTR and RTL Betti curves become zero for $x > 0.1$ because each vessel segment has terminated by $x = 0.1$. The $\beta_0(K^{\text{BTT}})$ and $\beta_1(K^{\text{BTT}})$ descriptor vectors increase steadily with $y$ as the horizontal plane moves past each individual vessel segment. The flooding descriptor vectors initially decrease with each flooding step and remain fixed at $\beta_0 = 1$, $\beta_1 = 0$ after six steps of flooding.

For the chemotaxis and haptotaxis-driven simulation (Figs 10D, 11E and 11F), $\beta_0(K^{\text{LTR}})$ decreases with $x$ as vessel segments anastomose together, while $\beta_1(K^{\text{LTR}})$ increases with $x$ due to the formation of many loops. The $\beta_0(K^{\text{RTL}})$ descriptor vector increases with $x$ until it reaches its maximum value of $\beta_0 = 11$ at $x = 0.85$ and then decreases to $\beta_0 = 0$. The $\beta_1(K^{\text{RTL}})$ descriptor vector decreases with $x$. The $\beta_0(K^{\text{BTT}})$ descriptor vector periodically increases and decreases with $y$ as the horizontal plane moves past regions of high and low connectivity. The $\beta_1(K^{\text{BTT}})$ descriptor vector steadily increases with the $y$ coordinate. The $\beta_0(K^{\text{flood}})$ descriptor vector is equal to one for all steps of flooding, and $\beta_1(K^{\text{flood}})$ decreases with each round of flooding and reaches $\beta_1 = 0$ after 20 flooding steps.

## Sweeping plane descriptor vectors cluster simulations by parameter values

We applied $k$-means clustering separately to the standard descriptor vectors and the topological descriptor vectors. None of the clusterings of the standard descriptor vectors were biologically interpretable in $(\rho, \chi)$ parameter space (see Fig 12 and S1 Fig). While the topological descriptor vectors from the sweeping plane filtration gave more biologically interpretable groupings for combined haptotaxis and chemotaxis parameter values, they were not robust for haptotaxis dominated or chemotaxis dominated $(\rho, \chi)$ parameter ranges (see S2 Fig). The flooding filtration did not give biologically interpretable groups for any parameter ranges (see **Clustering results** in S1 Appendix for all details). We suggest that the LTR and RTL plane sweeping filtrations produce the most interpretable clusterings because sweeping the plane horizontally resembles the way in which endothelial tip cells migrate from left to right in the simulated data.

No single descriptor vector was able to robustly cluster simulations into biologically interpretable groups. However, *double* descriptor vectors, created by concatenating two sweeping plane topological descriptor vectors, produced clusterings in $(\rho, \chi)$ parameter space that were biologically interpretable and robust. As an example of a double descriptor vector, "PIO$_0(K^{\text{LTR}})$ & PIO$_1(K^{\text{LTR}})$" denotes the topological descriptor vector created by concatenating the PIO$_0(K^{\text{LTR}})$ and PIO$_1(K^{\text{LTR}})$ descriptor vectors. We note that doubles of standard descriptor vectors or topological flooding descriptor vectors were unable to produce biologically interpretable and robust clusterings in $(\rho, \chi)$ parameter space (see **Clustering results** in S1 Appendix).

We computed a total of $2 \times 4 \times 3 = 24$ individual sweeping plane descriptor vectors for each model simulation: each descriptor vector counted connected components or loops $(\beta_0, \beta_1)$, swept a plane across the domain in one of four directions (LTR, RTL, TTB, and BTT), and used one of the three PH summaries (BC, PIO, PIR). We computed all double sweeping plane descriptor vectors and clustered all 276 combinations in $(\rho, \chi)$ parameter space. The 15 double descriptor vectors with the highest OOS accuracies are presented in Table 1; the four with the highest OOS accuracies are listed below:

1. PIO$_0(K^{LTR})$ & PIO$_0(K^{RTL})$ (93.1% OOS accuracy),

2. PIO$_0(K^{RTL})$ & PIR$_1(K^{LTR})$ (92.6% OOS accuracy),

3. PIR$_0(K^{TTB})$ & PIR$_1(K^{LTR})$ (91.5% OOS accuracy),

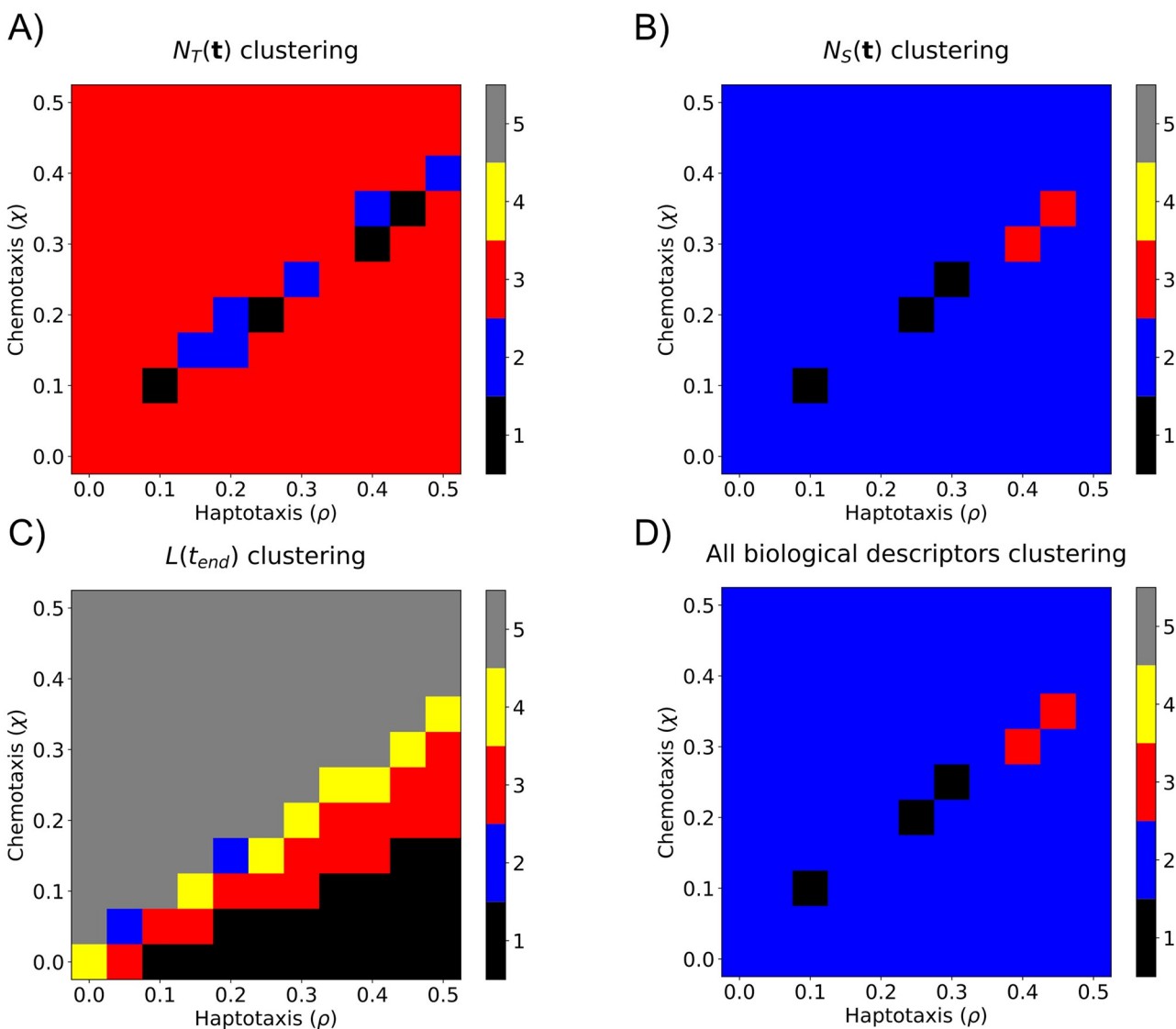

**Fig 12. Partitioning ($\rho, \chi$) parameter space into distinct regions using standard biological descriptor vectors.** We applied $k$-means clustering, with $k = 5$, to standard biological descriptor vectors. The resulting partitions are presented for A) $N_T(\mathbf{t})$, B) $N_S(\mathbf{t})$, C) $L(t_{end})$, and D) concatenation of all three descriptor vectors. The five clusters are ordered according to the mean $\chi$ value within the cluster.

4. $\text{PIR}_0(K^{BTT})$ & $\text{PIR}_1(K^{LTR})$ (91.5% OOS accuracy).

In Fig 13 and S3 Fig, we show how the data clustered in ($\rho, \chi$) parameter space when the top four double sweeping plane descriptor vectors were used. The "$\text{PIO}_0(K^{LTR})$ & $\text{PIO}_0(K^{RTL})$" and "$\text{PIO}_0(K^{RTL})$ & $\text{PIR}_1(K^{LTR})$" double descriptor vectors produced five connected clustering regions and, thus, attained high OOS accuracy and biologically interpretable clusterings. In contrast, the clusterings generated by the following double descriptor vectors ("$\text{PIR}_0(K^{TTB})$ & $\text{PIR}_1(K^{LTR})$" and "$\text{PIR}_0(K^{BTT})$ & $\text{PIR}_1(K^{LTR})$") were unable to distinguish between simulations with strong chemotaxis and strong haptotaxis. We conclude that double descriptor vectors comprising LTR and RTL descriptor vectors robustly cluster simulations into biologically interpretable clusterings because the associated filtrations capture the ways in which vessel

**Table 1. Summary of the 15 sweeping plane double descriptor vectors that achieved the highest OOS accuracy scores.**

| Feature | In Sample Accuracy | Out of Sample Accuracy |
|---|---|---|
| $PIO_0(K^{LTR})$ & $PIO_0(K^{RTL})$ | 94.0% | 93.1% |
| $PIO_0(K^{RTL})$ & $PIR_1(K^{LTR})$ | 94.2% | 92.6% |
| $PIR_0(K^{TTB})$ & $PIR_1(K^{LTR})$ | 92.3% | 91.5% |
| $PIR_0(K^{BTT})$ & $PIR_1(K^{LTR})$ | 92.3% | 91.5% |
| $PIR_0(K^{LTR})$ & $PIR_1(K^{LTR})$ | 92.3% | 90.9% |
| $PIO_0(K^{TTB})$ & $PIR_1(K^{LTR})$ | 94.2% | 90.4% |
| $PIO_0(K^{BTT})$ & $PIR_1(K^{LTR})$ | 94.3% | 90.1% |
| $PIO_0(K^{LTR})$ & $PIR_1(K^{LTR})$ | 92.6% | 90.1% |
| $PIR_0(K^{RTL})$ & $PIR_1(K^{LTR})$ | 92.6% | 89.0% |
| $PIR_0(K^{LTR})$ & $PIR_0(K^{RTL})$ | 90.2% | 89.0% |
| $PIO_0(K^{LTR})$ & $PIR_0(K^{RTL})$ | 91.9% | 88.4% |
| $PIO_0(K^{RTL})$ & $PIO_1(K^{LTR})$ | 86.1% | 86.0% |
| $PIO_0(K^{BTT})$ & $PIO_1(K^{LTR})$ | 84.2% | 85.1% |
| $PIO_1(K^{LTR})$ & $PIR_1(K^{BTT})$ | 85.6% | 84.8% |
| $PIO_1(K^{LTR})$ & $PIR_1(K^{TTB})$ | 83.1% | 84.8% |
| $PIO_0(K^{TTB})$ & $PIO_1(K^{LTR})$ | 84.9% | 84.8% |
| $PIO_0(K^{BTT})$ & $PIO_1(K^{RTL})$ | 85.6% | 84.8% |

segments anastomose and branch, respectively. We performed principal components analysis to reduce the "$PIO_0(K^{LTR})$ & $PIO_0(K^{RTL})$" double descriptor vector to two principal components. The results presented in S4 Fig show that the simulations from each group are also clustered in this two-dimensional space.

We applied $k-$means clustering to all sweeping plane double descriptor vectors and computed the average OOS accuracy for a range of values of $k$, also known as an elbow plot (S5 Fig). While the average OOS accuracy decreased with increasing values of $k$, the rate of decrease slowed markedly for $k > 5$. We fixed $k = 5$ as this value robustly clustered $(\rho, \chi)$ parameter space into biologically interpretable clusters (Fig 13). Larger values of $k$ may overfit the data [59], while smaller values fail to partition $(\rho, \chi)$ parameter space into biologically interpretable regions (S6 Fig). For example, when $k = 3$, simulations with strong haptotaxis ($\rho > \chi$), equal chemotaxis and haptotaxis ($\rho = \chi$), and strong chemotaxis ($\rho < \chi$) are clustered together. When $k = 4$, there are two distinct clusters near $\rho = \chi$, but simulations with large values of the haptotaxis parameter, $\rho$, are clustered with those with high values of the chemotaxis parameter, $\chi$. With $k = 5$ clusters, the region with large values of the haptotaxis parameter, $\rho$, was further subdivided. We suggest that clustering with $k = 5$ provides sufficient resolution to distinguish model simulations generated from different regions of parameter space. We note that more sophisticated descriptors could be used to justify this choice of $k$, including the silhouette score [60].

We identified five model simulations whose double descriptor vectors closely resembled the means from the clusters associated with the "$PIO_0(K^{LTR})$ & $PIO_0(K^{RTL})$" clustering. Fig 14 presents representative simulations of each cluster. We note that as $\rho$ and $\chi$ vary, the spatial extent and connectedness of the networks vary as expected (i.e., greater spatial extent in the $x$-direction as $\chi$ increases, and greater connectedness as $\rho$ increases). We applied a similar analysis to model simulations sampled from $(\rho, \chi, \psi)$ parameter space, where the non-negative parameter $\psi$ represents the rate at which tip cells branch. The resulting parameter clustering

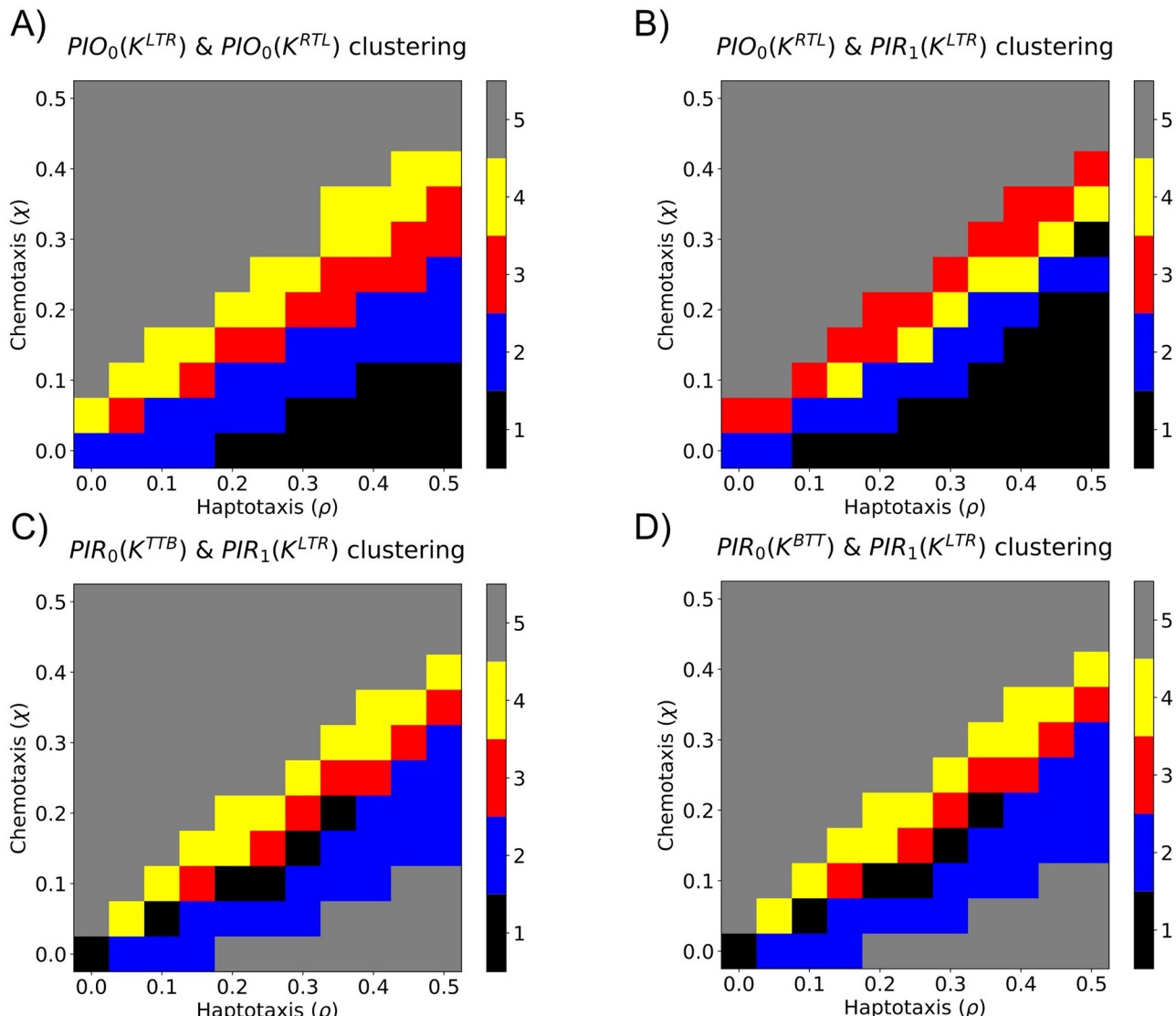

**Fig 13. Partitioning $(\rho, \chi)$ parameter space into distinct regions using sweeping plane double descriptor vectors.** We applied $k$-means clustering, with $k = 5$, to sweeping plane double descriptor vectors. The resulting partitions are presented for the four descriptor vectors that have the highest OOS accuracy: A) $PIO_0(K^{LTR})$ & $PIO_0(K^{RTL})$, B) $PIO_0(K^{RTL})$ & $PIR_1(K^{LTR})$, C) $PIO_0(K^{TTB})$ & $PIR_1(K^{LTR})$, and D) $PIO_0(K^{BTT})$ & $PIR_1(K^{LTR})$. The five clusters are ordered according to the mean $\chi$ value within the cluster.

changed only with $\rho$ and $\chi$, suggesting that the vascular morphology does not depend on $\psi$ (S7 Fig).

## Conclusions and discussion

We have introduced a topological pipeline to analyze simulated data of tumor-induced angiogenesis [17]. We investigated different filtrations to feed into the persistent homology algorithm. We demonstrated that simple data science algorithms can group together topologically similar simulated data. Pairs of topological descriptor vectors generated from sweeping plane filtrations robustly clustered simulations into five biologically interpretable groups.

## Mean Cluster Realizations from $PIO_0(K^{LTR})$ & $PIO_0(K^{RTL})$

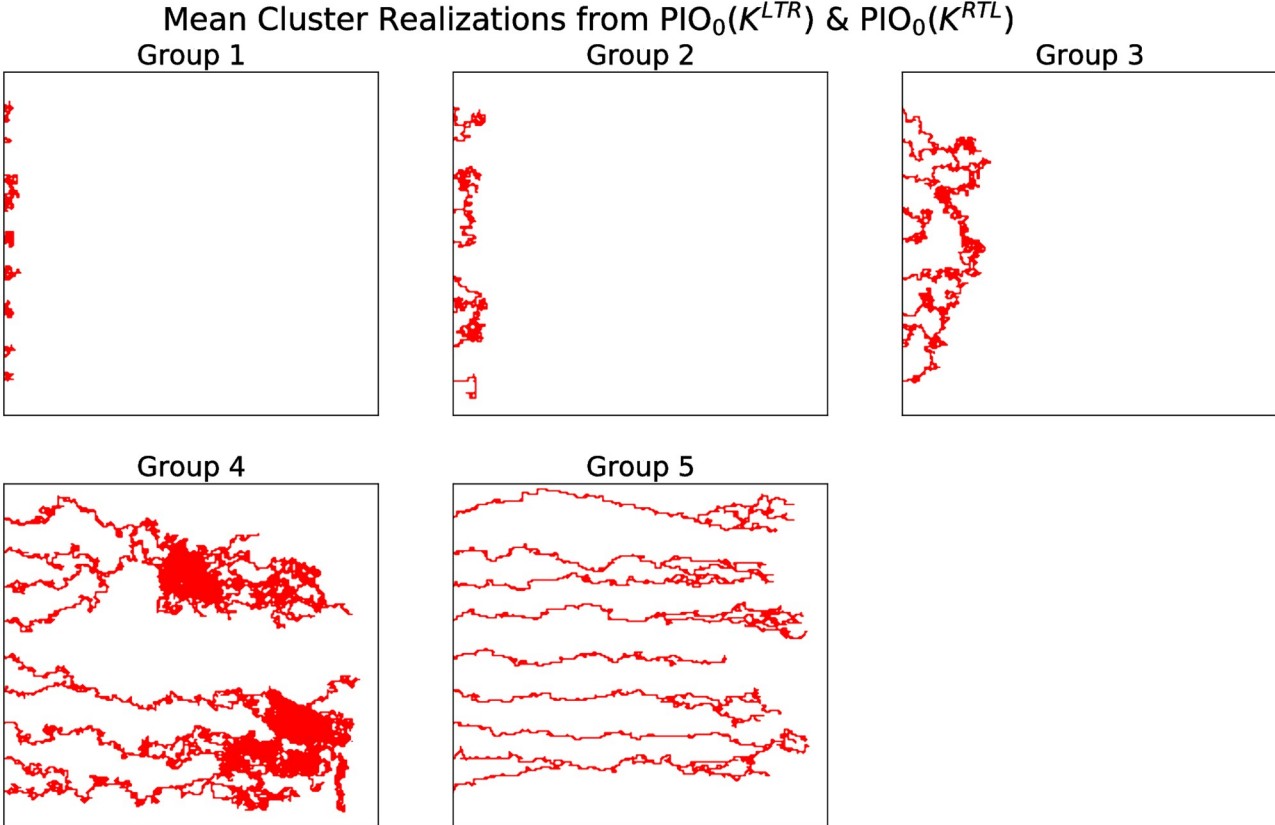

**Fig 14. Representative simulated blood vessel networks from Groups 1–5.** Series of simulations that resemble the means from each of the five groups (as identified from the double descriptor vectors associated with the sweeping plane filtrations $PIO_0(K^{LTR})$ & $PIO_0(K^{RTL})$). The representative simulations for groups 1–5 were generated using the following pairs of parameter values: $(\rho, \chi)$ = (0.25, 0), (0.5, 0.2), (0.3, 0.2), (0.15, 0.15), and (0.2, 0.4), respectively.

Simulations within each group were generated from similar values of the model parameters $\rho$ and $\chi$, which determine the sensitivity of the tip cells to haptotaxis and chemotaxis.

The Betti numbers quantify connectedness of the vasculature across spatial scales. We computed PH of four sweeping plane filtrations (LTR, RTL, TTB, BTT), which quantify topological features as a plane moves across the simulated data in a particular direction. We found that the filtration direction provides information about different aspects of the vascular network. For the LTR filtration, the number of connected components (i.e., $\beta_0$) decreases when an anastomosis event occurs, whereas for the RTL filtration, $\beta_0$ decreases when a branching event occurs. The TTB and BTT filtrations summarize vascular tortuosity: when tip cell movement in the $y$-spatial direction is negligible and/or tip cells do not merge with adjacent sprouts, $\beta_0$ increases with the number of tip cells/ number of sprouts. We found that the combined $PIO_0(K^{LTR})$ & $PIO_0(K^{RTL})$ sweeping plane descriptor vectors robustly cluster haptotaxis-chemotaxis parameter space into five distinct regions that appear to be connected.

Our results may depend on the model setup, in which new blood vessels are created as endothelial tip cells move from left to right. In the future, we may explore the use of extended persistence [61] and/or the PH transform [62], which allow consideration of multiple different directions in the topological analysis. If the vessel network grows radially, for example, then radial filtrations pinned at the network center may be more informative than the LTR filtration [45, 46]. Classically in PH, topological features of short persistence are topological noise, yet

there is growing evidence that these short features can provide information about geometry and curvature [43, 44, 46, 47, 63–66]. Our results using persistence images with unit weightings, rather than ramped weightings, also suggest that short persistence features may be informative for distinguishing vessel morphologies. The spatial resolution of the output binary images that we used for our analyses was the same as that used to generate the simulations, enabling us to distinguish the presence and absence of individual endothelial cells. In future work, we will consider finer and coarser binary images to investigate how the performance of the methods depends on their spatial resolution.

Many models of angiogenesis have been developed and implemented since the Anderson-Chaplain model, as has been extensively reviewed previously [21–24, 67–71]. We plan to use the methods described in this work to interpret more sophisticated models of angiogenesis and, in the longer term, to determine how vessel morphology affects tumor survival and to predict the efficacy of vascular targeting agents. The tumor is not explicitly modeled in the Anderson-Chaplain model, but many studies couple angiogenesis and tumor growth [25, 27, 29, 72, 73]. More complex models have been developed that account for mechanical factors such as mechanotaxis, pressure-driven convective transport of extracellular factors, mechanical stimulation of endothelial cell proliferation, and subcellular signalling pathways that guide cell fate specification of tip and stalk cells [26, 27, 29, 74]. For example, Vavourakis et al. developed a 3D model of angiogenic tumor growth that incorporates the effects of mechanotaxis on vessel sprouting and mechano-sensitive vascular remodelling [29]. Importantly, these more complex models have been shown to recapitulate experimental observations of time-varying spatial distributions of angiogenic vasculature [29] and features such as asymmetric neovasculature [74] and cell rearrangements and phenotype switching of endothelial stalk and tip cells [75, 76]. In future work, we will apply the TDA framework developed here to these, and other angiogenesis models. Such analyses will enable us to investigate the effect that processes such as mechanotaxis and subcellular signalling have on the topology of blood vessel networks.

Miller at al. used computational models to analyze tumor regression in response to drug treatment as a function of vascular morphology and heterogeneity [77]. Our future analysis of such state-of-the-art models will require us to extend the topological methods presented here for spatio-temporal data that describes how multiple cell types (e.g., tumor cells, immune cells and endothelial cells) interact in 3D. In a preliminary study, we applied these methods to 3D data from *in vivo* studies of tumor vessel networks, which is near the edge of current computational feasibility [46]. Further work in this direction requires significant computational advancements to scale the TDA pipeline to multiple, more complex models and to analyze the resulting parameter landscapes.

The topological approaches considered in this work represent a multiscale way to summarize the properties of *in silico* vasculature, and could be extended to experimental imaging data. Previous studies have quantified experimental data of tumor-induced angiogenesis using standard descriptors that include sprout length and diameter, number of tumor cells, concentration of angiogenic chemicals [30, 78, 79]. To compare models and data will require descriptors, such as the topological ones proposed here, that can discriminate between different conditions (e.g., parameter values). The challenges of making direct comparisons between experimental and synthetic data is a significant bottleneck in the validation of mathematical and computational models of angiogenesis. In future work we will apply the topological approaches outlined in this paper to compare different model simulations with experimental data [80, 81]. In this way, we aim to determine biologically realistic parameter values for control conditions [36], and to identify parameter values that are associated with response to treatments, including vessel normalization.

## Supporting information

**S1 Fig. Standard descriptor vector clustering.** Clustering of the $(\rho, \chi)$ parameter space using $k$-means with $k = 5$ on the standard descriptor vectors to summarize each simulated vasculature. We considered A) Number of tips over time, C) Number of vessel segments over time, E) Final length of the simulated vasculature, and G) All of the previous descriptor vectors concatenated into one descriptor vector. The five clusters are ordered according to the mean $\chi$ value within the cluster. Panels B,D,F,H depict the out of sample confusion matrices for each descriptor vector.
(TIF)

**S2 Fig. Individual plane sweeping descriptor vector clustering.** Clustering of the $(\rho, \chi)$ parameter space using $k$-means with $k = 5$ on individual sweeping plane topological filtrations to summarize each simulated vasculature. The four highest OOS accuracies resulted from the A) $PIR_1(K^{LTR})$, C) $PIO_0(K^{LTR})$, E) $PIO_1(K^{RTL})$, G) and $PIO_1(K^{LTR})$ descriptor vectors. The five clusters are ordered according to the mean $\chi$ value within the cluster. Panels B,D,F,H depict the out of sample confusion matrices for each descriptor vector.
(TIF)

**S3 Fig. Double plane sweeping descriptor vector clustering.** Clustering of the $(\rho, \chi)$ parameter space using $k$-means with $k = 5$ on double vectors of the sweeping plane topological filtration to summarize each simulated vasculature. The four highest OOS accuracies resulted from the A) $PIO_0(K^{LTR})$ & $PIO_0(K^{RTL})$, C) $PIO_0(K^{RTL})$ & $PIR_1(K^{LTR})$, E) $PIO_0(K^{TTB})$ & $PIR_1(K^{LTR})$, and F) $PIO_0(K^{BTT})$ & $PIR_1(K^{LTR})$ descriptor vectors. The five clusters are ordered according to the mean $\chi$ value within the cluster. Panels B,D,F,H depict the out of sample confusion matrices for each descriptor vector.
(TIF)

**S4 Fig. Dimensionality reduction.** Dimensionality reduction of the $PIO_0(K^{LTR})$ & $PIO_0(K^{RTL})$ double descriptor vector. We reduced the dimensionality of the $PIO_0(K^{LTR})$ & $PIO_0(K^{RTL})$ double descriptor vector to two dimensions using principal components analysis and plot the reduced-dimension descriptor vector for each simulation. The color of each dot denotes the predicted grouping of each simulation from the $k$-means algorithm.
(TIF)

**S5 Fig. Elbow curve.** Elbow curve for the OOS Accuracy for the $k$-means clustering and labeling algorithm over different values of $k$. We considered all 276 doubles of topological feature vectors and plot the mean OOS accuracy percentage plus or minus two standard deviations for multiple values of $k$. We propose that the elbow occurs at $k = 5$.
(TIF)

**S6 Fig. Multiple $k$-means clusterings.** Clustering of the $(\rho, \chi)$ parameter space using the $PIO_0(K^{LTR})$ & $PIO_0(K^{RTL})$ double of topological descriptor vectors using the $k$-means algorithm with A) $k = 3$, C) $k = 4$, E) $k = 5$. The clusters are ordered according to the mean $\chi$ value within the cluster. Panels B,D,F) depict the out of sample confusion matrices for each descriptor vectory.
(TIF)

**S7 Fig. $(\rho, \chi, \psi)$ clustering.** Clustering based on the $(\rho, \chi)$ grouping and the time to branch, $\psi$. We created a data set of blood vessel simulations by considering $(\rho, \chi)$ values from groups 1–5 from Fig 13A and letting $\psi$ vary over the 10 values $\{.1, 0.2, 0.3, \ldots, 1.0\}$. We simulated the Anderson-Chaplain model ten times at each $(\rho, \chi, \psi)$ values to create 500 total simulations. We

performed our clustering methodology on this collection of images based off the $PIO_0(K^{LTR})$ & $PIO_0(K^{RTL})$ descriptor vector.
(TIF)

**S8 Fig. Initial TAF and fibronectin profiles.** TAF concentration (left) and fibronectin concentration (right).
(TIF)

**S9 Fig. Individual flooding descriptor vector clustering.** Clustering of the $(\rho, \chi)$ parameter space using $k$-means with $k = 5$ on individual flooding topological descriptor vectors to summarize each simulated vasculature. The four highest OOS accuracies resulted from the A) $PIR_1(K^{flood})$, C) $\beta_1(K^{flood})$, E) $PIO_1(K^{flood})$, G) and $\beta_0(K^{flood})$ descriptor vectors. The five clusters are ordered according to the mean $\chi$ value within the cluster. Panels B,D,F,H depict the out of sample confusion matrices for each descriptor vector.
(TIF)

**S10 Fig. Double flooding descriptor vector clustering.** Clustering of the $(\rho, \chi)$ parameter space using $k$-means with $k = 5$ on double flooding topological descriptor vectors to summarize each simulation. The four highest OOS accuracies resulted from the A) $PIO_0(K^{flood})$ & $PIR_1(K^{flood})$, C) $PIR_0(K^{flood})$&$PIR_1(K^{flood})$, E) $\beta_0(K^{flood})$&$\beta_1(K^{flood})$, G) and $PIO_0(K^{flood})$ &$\beta_1(K^{flood})$ descriptor vectors. The five clusters are ordered according to the mean $\chi$ value within the cluster. Panels B,D,F,H depict the out of sample confusion matrices for each descriptor vector.
(TIF)

**S1 Table. Anderson-Chaplain model parameters.** List of mechanistic parameters for the Chaplain Anderson ABM from [17].
(PDF)

**S2 Table. Anderson-Chaplain model nondimensionalized parameters.** Baseline nondimensionalized mechanistic parameters for the Chaplain-Anderson Model from [17].
(PDF)

**S3 Table. Individual plane sweeping clustering.** Out of Sample Accuracy scores for individual feature vectors from the flooding filtration using $k$-means classification with $k = 5$.
(PDF)

**S4 Table. Double flooding clustering.** Out of Sample Accuracy scores for doubles of feature vectors from the flooding filtration using $k$-means classification with $k = 5$.
(PDF)

**S5 Table. Individual plane sweeping clustering.** Out of Sample Accuracy scores for individual feature vectors from the sweeping plane filtration using $k$-means classification with $k = 5$.
(PDF)

**S1 Appendix. Modeling and clustering appendix.** The file contains a more detailed description of implementation of the Anderson-Chaplain Model and clustering results.
(PDF)

## Author Contributions

**Conceptualization:** John T. Nardini, Bernadette J. Stolz, Kevin B. Flores, Heather A. Harrington, Helen M. Byrne.

**Formal analysis:** John T. Nardini, Bernadette J. Stolz, Kevin B. Flores, Heather A. Harrington, Helen M. Byrne.

**Funding acquisition:** Kevin B. Flores, Heather A. Harrington, Helen M. Byrne.

**Investigation:** John T. Nardini, Bernadette J. Stolz, Kevin B. Flores, Heather A. Harrington, Helen M. Byrne.

**Methodology:** John T. Nardini, Bernadette J. Stolz, Kevin B. Flores, Heather A. Harrington, Helen M. Byrne.

**Project administration:** Kevin B. Flores, Heather A. Harrington, Helen M. Byrne.

**Software:** John T. Nardini.

**Supervision:** Kevin B. Flores, Heather A. Harrington, Helen M. Byrne.

**Validation:** John T. Nardini, Bernadette J. Stolz, Kevin B. Flores, Heather A. Harrington, Helen M. Byrne.

**Visualization:** John T. Nardini, Bernadette J. Stolz, Kevin B. Flores, Heather A. Harrington, Helen M. Byrne.

**Writing – original draft:** John T. Nardini, Bernadette J. Stolz, Kevin B. Flores, Heather A. Harrington, Helen M. Byrne.

**Writing – review & editing:** John T. Nardini, Bernadette J. Stolz, Kevin B. Flores, Heather A. Harrington, Helen M. Byrne.

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
