## [Decision Letter · Decision Letter 0]

16 Feb 2021

Dear Professor Byrne,

Thank you very much for submitting your manuscript "Topological data analysis distinguishes parameter regimes in the Anderson-Chaplain model of angiogenesis" for consideration at PLOS Computational Biology.

As with all papers reviewed by the journal, your manuscript was reviewed by members of the editorial board and by several independent reviewers. In light of the reviews (below this email), we would like to invite the resubmission of a significantly-revised version that takes into account the reviewers' comments.

Dear Authors,

Your manuscript has been reviewed by three experienced reviewers and each has recommended that revisions be undertaken before the paper can be considered acceptable for publication.

Please revise your manuscript in line with the comments of each of the reviewers and provide details of the changes made when re-submitting the revised version.

We look forward to receiving your revised manuscript in due course which will be sent back to the reviewers.

We cannot make any decision about publication until we have seen the revised manuscript and your response to the reviewers' comments. Your revised manuscript is also likely to be sent to reviewers for further evaluation.

Sincerely,

Mark Chaplain

Guest Editor

PLOS Computational Biology

Daniel Beard

Deputy Editor

PLOS Computational Biology

Dear Authors,

Your manuscript has been reviewed by three experienced reviewers and each has recommended that revisions be undertaken before the paper can be considered acceptable for publication.

Please revise your manuscript in line with the comments of each of the reviewers and provide details of the changes made when re-submitting the revised version.

We look forward to receiving your revised manuscript in due course which will be sent back to the reviewers.

Reviewer's Responses to Questions

**Comments to the Authors:**

Reviewer #1: This paper proposes a topological data analysis (TDA) pipeline to interrogate simulation predictions of the Anderson-Chaplain model of angiogenesis associated with tumour development. The authors assess various topological and standard descriptors of model simulations generated by different parameter values, and they demonstrate TDA’s performance to stratify the A-C model parameter space into regions with regards to vessel morphology similarities. The paper is in general very comprehensive and structured. A well-executed study is presented here that provides important results pertinent for systematic analysis of the A-C model, and with regards to a specific TAD algorhtm: persistent homology. The reviewer recommends publication once the following revisions have been addressed by the authors. Please note that the comments for revision are not presented below in any special order.

Introduction (2nd paragraph): Here the authors outline the A-C angiogenesis model, which appears not to account for any vascular remodelling (vessel compression, collapse, lumen remodelling, etc.) - not as far as more contemporary models do, e.g.: DOI:10.1016/j.mvr.2015.02.007, DOI:10.1371/journal.pone.0150296, DOI:10.1371/journal.pcbi.1005259, DOI:10.1098/rsif.2016.0918. Therefore, the authors should justify this simplification in their study. Also, they need to elaborate in the paper how and if this would impact the findings of this work. Along the same lines, the authors should justify why they decided to select investigating the A-C model using a two-dimensional space. They have to elaborate and convince on their decision to do so – mainstream state-of-art mathematical models are 3D; thus, a 2D cancer model study may seem too simplistic nowadays. The authors need to elaborate on this matter and explain again if the findings of their TDA study would be different in the case of a A-C model in 3D.

Introduction (2nd paragraph) + Discussion: The review articles of theoretical cancer models should include more recent ones, opposed to the ones currently cited [41-44], more specifically the reviewer proposes: DOI:10.1200/CCI.18.00068, DOI:10.1088/1478-3975/ab1a09, DOI:10.1016/j.ymeth.2020.02.010.

Figure 1: This figure is nice despite no clear explanation why/how fibronectin gradients are produced, while also the vascular elements recruited by TAFs that are in turn secreted by tumour cells seem too out of scale compared to the “mother” blood vessel. Minor rework is needed here to further improve the cartoon.

Figure 2: Here the authors outline the hierarchy of the proposed pipeline, where the 3 major components (mathematical model -> data analysis -> data clustering) are shown very nicely. However, one may say that box 2 and box 3 concern about data processing and analysis; therefore, there is no clear distinction between the two. Also, there is no connection amongst the 3 boxes that is to say which component from box 1 feeds another component from box 2, and which component from box 2 supplies box 3 or/and box 1, and so on. The illustration needs some rework to render it a true figure showing the “flow” of the processes involved in the proposed pipeline!

Materials & Methods: The major direction of tumour angiogenesis data via a topological data analysis of the Anderson-Chaplain model is to interrogate haptotaxis and chemotaxis. However, the papers cited already in this manuscript [47,48] highlight the importance of mechanotaxis in tumour angiogenesis. This is a major simplification which may also have significant impact to the TDA analysis of the tumour networks. The authors need to elaborate on this in the paper, and they are strongly advised to encompass in their study the effect of mechanotaxis.

Materials & Methods: “We generate 10 realizations of the model for each of the 121 parameter combinations, and produce… “ It should be justified how this parameter space discretization and why this parameter range was selected. Should the authors have selected a much broader or narrower parameter range for model parameters ‘rho’ and ‘chi’, or even a parameter range that may produce “unrealistic simulations,” would this have affected the findings significantly, or not? Please elaborate on this – the authors should also connect their response with the above reviewer comment!

Materials & Methods: “To facilitate downstream clustering analysis, we require vectors of the same length for all simulations.” The authors need to justify this decision. What if the ‘experimental data’ were not homogeneous and of a different sample size. Would this impact the reliability and accuracy of the proposed pipeline; would the pipeline be able to work in principle? Please explain that in the manuscript.

Figure 3: Selection of vessel segments (that would in principle determine the number of vessel segments) appears somehow arbitrary. Please explain how it works in the algorithm. Also, the concept determining the final length of the vessels may be appropriate for a 2D representation of a vasculature. Despite of this, what if one had considered to adopt a different wave-front determination algorithm for the tip vessel propagation “monitoring.” What if one had employed in the algorithm multiple wave-fronts / directions of the dashed-line(s). Would the proposed approach work in 3D representation of the tumour vessels? Please elaborate further details of the algorithm in the manuscript.

Filtrations for simulated vasculature / Figure 4: It is not entirely clear in the methods how the binary image is generated – here from the model produced data (synthetic data) – and how it is associated with the resolution of the simulation space. More precisely, the authors need to explain how the “resolution” of the model generated tumour vasculature relates to the binary image “resolution”. Is there any relationship between the two resolutions? For example, let’s take two extreme scenarios: (a) in the first one, assume the resolution of the model vasculature (produced using the snail-trail algorithm) is of the order of 1 [unit length] (minimum segment length) whereas the pixel size is 10*10 [unit lengtht]^2; (b) in the second scenario, the resolution of the model vasculature is of the order of 1 [unit length] (minimum segment length) whereas the pixel size is 1*1 [unit lengtht]^2. Can the authors explain how the filtration algorithm will perform – please quantify your results.

The authors need to comment also about the efficiency of the “sweeping plane filtration” and “flooding filtration” algorithms when applied to 3D data (i.e., image stacks of MR scans, see review DOI:10.1018/nrclinonc.2014.126).

The following statement is not entirely clear to the reviewer, please consider rephrasing it.

“On each subsequent step, we iterate through each nonzero pixel from the previous binary image and manually set every pixel in its Moore neighborhood …”

“From the corresponding sequence of binary images, we construct a filtered simplicial complex K^flood = {K_1, K_2, …, K_end}.” To the reviewer’s understanding, end filtration will result into a simplicial complex that covers the entire “image”. Can the authors explain why this is needed? Is this right?

Pages 8-9: “Intuitively, a topological feature is born in filtration step … with another component or when a loop is covered by 2-simplices.”

It is self-evident that the homology group for when a feature is generated / born (tip vessel, branch, anastomosis, etc.) and for when a feature ceases to exist (e.g., tip transforms into an anastomosis). Nonetheless, and in view of the modelling simplification that no vascular remodelling / vessel compression or collapse / vessel intussusception is encompassed in the A-C model, then how a branch (or even a tip vessel) is regressed, or how an anastomosis is pruned in the vascular network.

Topological descriptor vectors from PH: The following statement may need some further justification why this step was considered in the persistent homology algorithm.

“In a first step the coordinates (b; d - b) are blurred by a Gaussian, with standard deviation sigma, that is centered about each birth-persistence point.”

Simulation Clustering: “Specifically, 7 of the 10 descriptor vectors from each of the 121 (rho, chi) parameter combinations …” were these 7 descriptor vectors begin selected randomly following a specific pattern (uniform, normal distribution)?

Results / Haptotaxis and chemotaxis alter vessel morphology: As covered in the recent reviews (DOI:10.1007/s11831-015-9156-x, DOI:10.1016/j.ymeth.2020.02.010 - they should be cited here), angiogenesis models have already highlighted the importance of chemotaxis and haptotaxis in tumour-induced angiogenesis, whereas in these two modelling papers (DOI: 10.1371/journal.pcbi.1005259) and (DOI:10.1098/rsif.2018.0415) it has been demonstrated the importance of mechanotaxis and interstitial fluid flow in vascular development in the tumour microenvironment. Therefore, the contribution of this paragraph is not entirely clear to the reviewer. It should be enhanced with the analysis suggested in the comment #5 above.

Reviewer #2: Topological data analysis distinguishes parameter regimes in the Anderson-Chaplain model of angiogenesis

The authors present a novel analysis pipeline that applies multi-scale encoding and machine learning methods to simulated data produced by a well known model of cancer-induced angiogenesis, in order to identify biologically similar groupings of vessel morphology. The idea is well founded and the motivation clear, and the methods are sound. The paper is also generally well written and easy to digest.

However, I think some major clarification is necessary to the results, as it is very unclear as to why one particular encoding (the sweeping plane filtration) produces better results than another (the flooding filtration). Furthermore, this comparison is confused with the overall message, which (I think) is that a standard clustering method (k-means) can be used with multi-scale encodings to group similar vascular morphologies, while standard post-processed metrics (e.g. number of nodes, vessel length) cannot. I think the authors need to clearly separate the two questions.

General comment

The authors frequently make reference to “biologically interpretable / informative / meaningful” clusters. However the definition of “biologically meaningful” is not clear to me; I would guess it refers to clusters that have similar biological properties, but this is purely my inferred meaning. I think the authors should define this up front with a clear statement on what constitutes a “biologically meaningful cluster”. Furthermore, there’s a difference between “meaningful” and “interpretable”; the latter term has a quite specific meaning in machine learning that is not necessarily equivalent to meaning (an algorithm can be non-interpretable but still produce meaningful predictions).

Materials and methods

Have the authors considered alternative standard descriptors from the literature? For example, δmax, the maximum distance in the tissue from the nearest blood vessel, and λ, a measure of the shape of the spaces between vessels, used by https://www.pnas.org/content/108/5/1799. Or even simply tortuosity (which the authors themselves reference on page 13).

Section ‘Filtrations for simulated vasculature’. The results section raises questions regarding the two representations presented here (sweeping plane and flooding). Given that they produce quite different results, I think some reference should be made to previous applications of each encoding; in particular what problems they were developed to solve, as this may shed light on why sweeping plane performs better than flooding.

Section ‘Simulation clustering’. The distance metric used by the k-means algorithm is not explained; I assume it’s Euclidean, but the authors should state this clearly.

Results

Figs 10, 11. I find it difficult to compare these figures because the Betti curves are presented on different axes (space in Fig 10, step in Fig 11). It would help if the figures were more directly comparable, or even on the same plot.

Section ‘Haptotaxis and chemotaxis alter vessel morphology’. The respective roles of hapto- and chemotaxis have been investigated before, it’s not entirely clear whether the authors are reporting them as new or just including them for context. I would guess the latter, but I think clarification is needed.

Section ‘Sweeping plane descriptor vectors cluster simulations by parameter values’. There are two results here that are not explained: i) sweeping plane can produce biologically meaningful clusters, while flooding cannot; ii) double descriptor vectors created by sweeping plane produce biologically meaningful and robust clusters. I think the authors need to provide explanations, and in particular comment on why the two encodings produce such different results.

Choosing the value of k: the elbow plot (SI Fig 15) is not hugely convincing to justify k=5, and the decision appears to have been made somewhat post hoc based on the desired interpretation of the clusters. The silhouette score might provide more information, and should be relatively easy to calculate given that a standard distance metric is used by the k-means algorithm.

Reviewer #3: Nardini and colleagues present an application of topological data analysis (TDA) to analyse synthetic vascular network generated with the Anderson-Chaplain (A-C) model of tumour angiogenesis. The authors compare various choices of description vector arising from the TDA and demonstrate that a judicious choice enables robust clustering of the synthetic networks based on the relative weight given to chemo-/haptotaxis in a number of simulated networks even in the presence of stochasticity.

There is currently a lack of methods for defining embeddings of vascular networks that enable downstream classification and regression. This is particularly challenging in this scenario given the intrinsic structural stochasticity. This is a very substantial contribution of this study and I congratulate the authors on the innovation provided. The writing in the manuscript is impeccable and the figures provided are of high quality, which facilitates substantially the understanding of the concepts introduced and discussed.

Before the manuscript can be accepted, I would like to request the following corrections:

1. At the risk of sounding pedantic, the description of tumour-induced angiogenesis in Fig. 1 needs its language updated or simply make clear that some of the concepts are useful modelling abstractions but not necessarily biological mechanisms. For example, endothelial cells (ECs) don’t “deposit” other ECs as they invade the avascular space. Since the formulation of the original A-C model, there have been substantial advances in understanding the dynamics of tip and stalk cells in sprouting angiogenesis. Furthermore, ECs don’t branch into two cells, it is ECs along an existing sprout that initiate secondary sprouts and so on. It may be worth bringing the description of TAF (VEGF+bFGF) from the methods section to the main text.

2. The argument of the paper goes that the standard quantitative descriptors in page 4 (inter-vessel spacing, number of branching points, tortuosity, etc) fail to capture higher dimensional information and facilitate network clustering. It would be very interesting to see how (little) effective these are at clustering the simulations similarly to Fig. 12. This comparison would provide quantitative evidence of the superior performance of the method proposed.

3. Fig. 12 and accompanying text: it would be interesting to also see the PIO0(KLTR)& PIO0(KRTL) descriptors plotted on a 2D plane (via Principal Component Analysis for example) to get a sense of the distance between the clusters. Fig. 13 appears to indicate that classes 1 and 2 may be phenotypically much closer than any other pair.

4. Page 18: my main criticism of the study is that the conclusion that the proposed descriptors cluster the networks in biologically meaningful way is highly dependent on the choice of k=5 in the clustering analysis. This choice is justified in a very empirical way and I would suggest the authors to use the elbow method or a similar heuristic to argue about the optimal number of clusters in k-means based on the amount variance explained.

**Have all data underlying the figures and results presented in the manuscript been provided?**

Reviewer #1: Yes

Reviewer #2: Yes

Reviewer #3: Yes

PLOS authors have the option to publish the peer review history of their article (what does this mean?). If published, this will include your full peer review and any attached files.

Reviewer #1: No

Reviewer #2: No

Reviewer #3: No
---

## [Decision Letter · Decision Letter 1]

18 May 2021

Dear Professor Byrne,

We are pleased to inform you that your manuscript 'Topological data analysis distinguishes parameter regimes in the Anderson-Chaplain model of angiogenesis' has been provisionally accepted for publication in PLOS Computational Biology.

Best regards,

Mark Chaplain

Guest Editor

PLOS Computational Biology

Daniel Beard

Deputy Editor

PLOS Computational Biology

Thank you for revising the paper and taking into account all comments of the reviewers.

#Accept

Reviewer's Responses to Questions

**Comments to the Authors:**

Reviewer #1: The authors have provided a detailed and thorough response/revision to my comments - I endorse the paper for publication.

Reviewer #3: The authors have addressed my comments satisfactorily.

**Have the authors made all data and (if applicable) computational code underlying the findings in their manuscript fully available?**

Reviewer #1: **No: **The code for the Chaplain Anderson ABM model used in this study should be provided. Input files to run the code should be also provided to replicate the results of this study.

Reviewer #3: Yes

PLOS authors have the option to publish the peer review history of their article (what does this mean?). If published, this will include your full peer review and any attached files.

Reviewer #1: **Yes: **Dr Vasileios Vavourakis

Reviewer #3: No

---

## [Editor Report · Acceptance letter]

25 Jun 2021

PCOMPBIOL-D-21-00052R1 

Topological data analysis distinguishes parameter regimes in the Anderson-Chaplain model of angiogenesis

Dear Dr Byrne,

I am pleased to inform you that your manuscript has been formally accepted for publication in PLOS Computational Biology. Your manuscript is now with our production department and you will be notified of the publication date in due course.

With kind regards,

Katalin Szabo
